# Efficient spatially targeted gene editing using a near-infrared activatable protein-conjugated nanoparticle for brain applications

Catarina Rebelo[1,2], Tiago Reis[1], Joana Guedes [1], Cláudia Saraiva[3], Artur Filipe Rodrigues [1], Susana Simões[1], Liliana Bernardino[3], João Peça[1,4], Sónia L. C. Pinho [1✉] & Lino Ferreira [1,2✉]

Spatial control of gene expression is critical to modulate cellular functions and deconstruct the function of individual genes in biological processes. Light-responsive gene-editing formulations have been recently developed; however, they have shown limited applicability in vivo due to poor tissue penetration, limited cellular transfection and the difficulty in evaluating the activity of the edited cells. Here, we report a formulation composed of upconversion nanoparticles conjugated with Cre recombinase enzyme through a photo-cleavable linker, and a lysosomotropic agent that facilitates endolysosomal escape. This formulation allows in vitro spatial control in gene editing after activation with near-infrared light. We further demonstrate the potential of this formulation in vivo through three different paradigms: (i) gene editing in neurogenic niches, (ii) gene editing in the ventral tegmental area to facilitate monitoring of edited cells by precise optogenetic control of reward and reinforcement, and (iii) gene editing in a localized brain region via a noninvasive administration route (i.e., intranasal).

[1] CNC-Center for Neurosciences and Cell Biology, University of Coimbra, Coimbra, Portugal. [2] Faculty of Medicine, University of Coimbra, Coimbra, Portugal. [3] Health Sciences Research Centre (CICS-UBI), University of Beira Interior, Covilhã, Portugal. [4] Department of Life Sciences, University of Coimbra, Coimbra, Portugal. ✉email: slpinho@uc-biotech.pt; lino@uc-biotech.pt

Tackling complex biological questions, such as dissecting neuronal circuit function or stimulating regenerative processes in an intact organism, requires sophisticated technologies that facilitate precise genetic modification in live animals. Several approaches have been recently developed to control the activity of genes with precise spatiotemporal control, using enzymes such as Cre recombinase[1–3] or Cas9[4–6] that can be externally triggered by light. Nevertheless, despite the great progress made in recent years, our ability to controlling gene expression within biological tissue with efficiency and spatial resolution, and simultaneously evaluate the activity of the edited cells by optogenetics has not been demonstrated. Viral strategies used to deliver genes or gene-modifying cargo into the mammalian brain are limited by several drawbacks including: (i) low biosafety profile of several commonly used vectors, (ii) limitations in cargo size capacity, and (iii) nonspecific infection of pre- and postsynaptic neurons at injection sites. Moreover, our capacity to noninvasively deliver gene-editing formulations into the brain is very limited. Most non-viral gene-editing formulations have been administered via an intracerebral route (reviewed in ref. [7]). Whether nonviral formulations can be administered through a non-intracerebral route and accumulate in the brain remains to be investigated. This strategy could enormously simplify the application of gene-editing formulations in the brain.

Regulation of Cre recombinase formulations with light exposure has been shown to provide control over biological processes with an unprecedented precision. Light-activatable Cre recombinase systems have been based in split-Cre dimerization[2,3,8–10], the installation of a light-responsive o-nitrobenzyl caging group in the catalytic site of Cre[11] or via the fusion of a photocleavable protein to the Cre recombinase[1]. However, these systems still display considerable limitations: (i) they are based on the constitutive expression of Cre, and (ii) they rely on the activation using UV/blue light, which has limited tissue penetration ability. A recent study has attempted to address these issues by developing a nanoparticle formulation capable of delivering Cre recombinase after activation by near-infrared (NIR) light[12]. However, this approach relied on the immobilization of Cre by noncovalent interactions with gold nanoshells, which resulted in poor control over protein delivery. In addition, the formulation showed limitations in terms of cellular uptake and intracellular delivery, which together translated into low cell recombination in vitro (approximately 20%), and its gene-editing potential was not demonstrated in vivo.

Here, we report the in vivo transcranial delivery of a protein-conjugated formulation activated by NIR light. This system is composed of lanthanide-doped upconversion nanocrystals (UCNPs) capable of converting NIR light into blue light, a photocleavable linker (PCL) attaching Cre to the UCNPs, and surface modification of the UCNPs with hydroxychloroquine (HCQ) to facilitate endolysosomal escape (Fig. 1). UCNPs have been investigated for the release of small molecules[13–15], some into the brain, and for their use as optogenetic actuators of transcranial NIR light to stimulate/inhibit deep brain nuclei[16,17]. However, to date, no formulation has achieved in vivo NIR-activated Cre recombinase-mediated gene editing with spatial resolution. In this work, the activity of the edited cells was evaluated by optogenetics, using blue light as a trigger. We show that stereotaxic administration of UCNPs conjugated with Cre recombinase (Cre-UCNPs) in the subventricular zone (SVZ) region led to the recombination of a loxP cassette in brain cells of transgenic mice only after exposure to a continuous-wave NIR laser. We further show that the stereotaxic administration of the Cre-UCNPs in the ventral tegmental area (VTA) followed by NIR activation allowed us to gain precise control over reward processes and achieve behavioral reinforcement via blue

light Cre-dependent channelrhodopsin-2 (ChR2) activation[16,18]. Finally, we show spatial resolution after gene editing mediated through Cre-UCNPs delivered via a non-invasive administration route (i.e., intranasal instillation).

## Results

**Preparation and characterization of light-triggered Cre-UCNPs.** Core-shell, lanthanide-doped NaYF$_4$:Tm,Yb UCNPs (Fig. 1) were selected for the design of an NIR activatable editing system and synthesized according to established protocols[19]. Yb$^{3+}$ acted as the sensitizer, enabling efficient NIR absorption in the range of 900-1000 nm, while Tm$^{3+}$ acted as the activator ion, enabling blue- to UV-shifted emissions[19]. Incorporation of the NaYF$_4$ shell improved the fluorescence intensity of the core UCNPs by minimizing quenching effects caused by interactions of dopant ions with the solvent[20]. Powder X-ray diffraction confirmed that the UCNPs were highly crystalline and hexagonal in phase (Supplementary Figs. 1a–b). The UCNPs also exhibited characteristic upconversion emission peaks at 345, 362, 450, and 475 nm after activation with a 980 nm NIR laser (Supplementary Fig. 1c). The conversion yield of NIR to blue light was ~2–4% (Supplementary Fig. 1d, e).

To allow the chemical immobilization of biomolecules, the surface of the UCNPs was coated with a mixture of amino- and azido-containing organosilanes, as silica-coated UCNPs have shown good biocompatibility with brain tissue in vivo[16]. The resulting N$_3$/NH$_2$-UCNPs showed an upconversion spectrum similar to that of nonsilanized samples (Supplementary Fig. 1c), and an average diameter of 25.6 ± 1.1 nm, as determined by transmission electron microscopy (TEM) analyses (Supplementary Fig. 1f, g). The presence of amine and azide groups was confirmed by Fourier transform infrared (FTIR) spectroscopy (Supplementary Fig. 2a), fluorescamine quantification (~830 surface amine groups per UCNP) and ζ-potential measurements (Supplementary Fig. 2d). The UCNP surface amine groups reacted with the hydroxyl groups in HCQ (Supplementary Fig. 2b). HCQ is a clinically approved drug used to treat malaria and lymphoma, and it has lysosomotropic properties contributing to the endosomal escape of NPs[21]. However, the effect of HCQ on the endosomal escape of protein cargo is largely unknown[22]. Approximately 60 HCQ molecules were conjugated to each UCNP, as determined by absorbance measurements[21]. The UCNP surface azide groups were reacted with the alkyne group of the PCL, which contained a terminal carboxyl group enabling the reaction with the amine groups of Cre recombinase (Fig. 1b and Supplementary Fig. 2c). The resultant formulation displayed an average diameter of 28.0 ± 2.8 nm as determined by scanning TEM analyses (Fig. 2a and Supplementary Fig. 1g), an upconversion spectrum that overlaps the absorbance of the PCL (Fig. 2b) and approximately 9.5 μg of Cre per mg of UCNP, corresponding to an average total of 4 proteins per UCNP (hereafter termed as Cre-UCNP). According to dynamic laser scattering studies, the Cre-UCNPs were negatively charged when suspended in culture medium and relatively stable for at least 48 h (Supplementary Fig. 3). Importantly, the Cre-UCNP formulation released approximately 58% of the immobilized protein after exposure to an NIR laser (980 nm, 785 mW/cm$^2$) for 2 min (Fig. 2c).

**Internalization, cytotoxicity and intracellular trafficking of light-triggered UCNPs.** The uptake of Cre-UCNPs by a fibroblast Cre reporter cell line was determined by inductive coupled plasma mass spectrometry (ICP-MS) (Fig. 2d). The cellular internalization of the Cre-UCNPs was relatively rapid (approximately 2-4 h) and was dependent on the NP concentration and

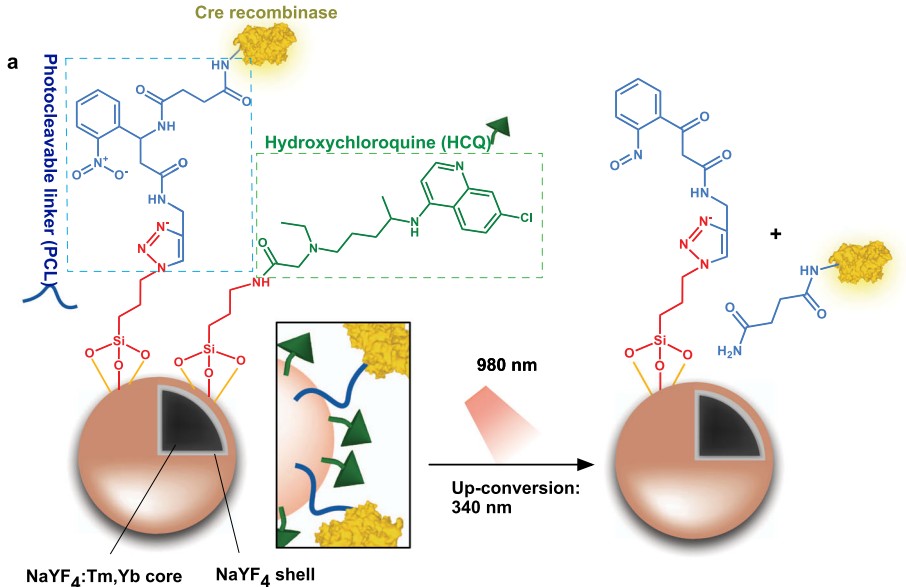

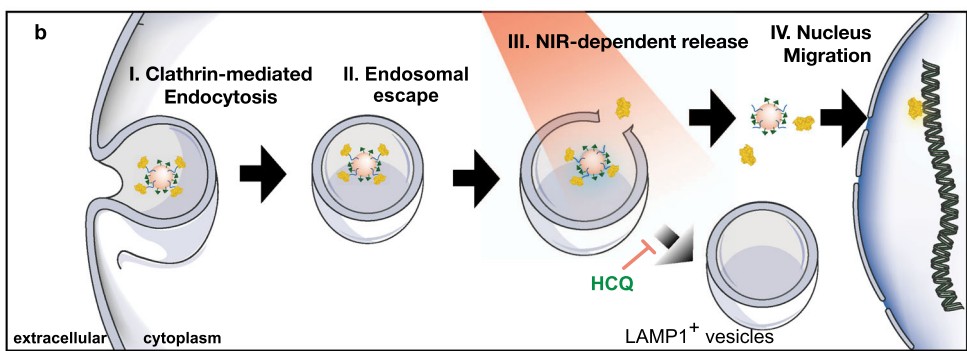

**Fig. 1 Composition and action mode of Cre-UCNPs. a** Schematic representation of Cre-UCNPs. Initially, the amine groups of silanized UCNPs were reacted with hydroxychloroquine (HCQ) while the azide groups were used for the immobilization of NLS-Cre recombinase through a photo-cleavable linker (PCL). NLS is a nuclear localization signal. Cre-UCNPs had approximately 60 molecules of HCQ and 4 molecules of Cre recombinase per NP. Upon irradiation with a 980 nm NIR light, the UCNP core emits light in the UV range. PCL is cleaved by the 340 nm radiation emitted from UCNP and releases the enzyme from the NP. **b** Gene edition mediated by Cre-UCNPs. Cre-UCNPs were internalized by cells through endocytosis. HCQ present in the NP has a lysosome inhibitory effect and assists on the release from the endosomal compartments. Upon NIR activation, Cre recombinase is released, diffuses to the nucleus with the help of NLS sequence and induces recombination. Adapted from images obtained from Servier Medical Art by Servier (http://smart.servier.com), licensed under a Creative Commons Attribution 3.0 Unported License, and from RCSB Protein Data Bank (doi: 10.2210/pdb1q3v/pdb) using the 3D View tool.

presence of the HCQ (Fig. 2d and Supplementary Fig. 4a). To evaluate the possible cytotoxic effects of the Cre-UCNPs and NIR laser irradiation (980 nm, 785 mW/cm$^2$), cells were exposed to a range of concentrations of Cre-UCNPs (from 25 to 500 μg/mL) and various light pulse durations (Supplementary Figs. 4b–g). No measurable cytotoxicity effect was observed for concentrations less than or equal to 50 μg/mL and irradiation times less than 30 min. Under these conditions, irradiated cells showed no heat shock response, which was monitored by heat shock protein 70 staining[23] (Supplementary Fig. 5), and no double-stranded DNA breaks were found, as determined by assessment of the phosphorylation of a histone H2A.X variant[24] (Supplementary Fig. 6).

To identify the pathway through which our negatively charged Cre-UCNP penetrated cells, we exposed fibroblasts to fluorescently labeled Cre-UCNPs in the presence of chemical or siRNA-mediated endocytosis inhibitors (Fig. 2e and Supplementary Fig. 7a–d) and assessed particle internalization via flow cytometry. Our chemical and siRNA inhibition results showed

that the main internalization mechanism was mediated by clathrin-mediated endocytosis and, to a lesser degree, by macropinocytosis. These cellular uptake pathways were also supported by TEM analyses (Supplementary Fig. 7e).

To evaluate the intracellular trafficking of the Cre-UCNPs, cells were exposed to Cre-UCNPs (50 μg/mL) and monitored by confocal microscopy (Fig. 2f) to determine the staining of Cre and lysosome-associated membrane protein 1 (LAMP1). After 5 min of exposure to NPs, the cells were characterized, and the co-localization of Cre-UCNPs with stained LAMP1 foci was reduced compared to that of the Cre-UCNPs without HCQ. Our results further showed that cells exposed to Cre-UCNPs with HCQ had a lower number of LAMP1-positive vesicles than those exposed to Cre-UCNPs without HCQ, which may indicate disrupted lysosome biogenesis (Supplementary Fig. 8). The decrease in the colocalization of Cre-UCNPs with LAMP1 may be explained by the fact that HCQ impairs autophagosome fusion with lysosomes[22], and thus preventing the accumulation of NPs in LAMP1-positive vesicles, and/or by the disruption of the

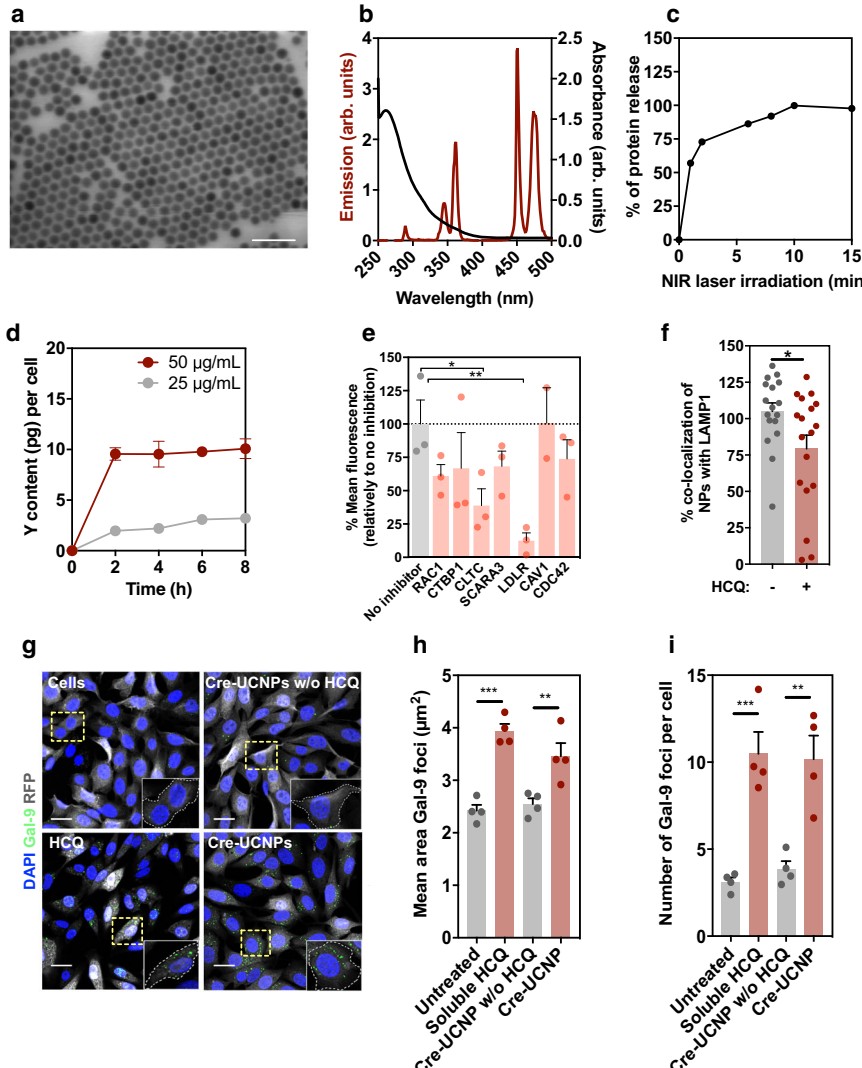

**Fig. 2 Cellular uptake and intracellular trafficking of Cre-UCNPs. a** Representative STEM image of Cre-UCNPs, obtained from a total of 5 images. Scale bar = 100 nm. **b** Luminescence emission spectrum of UCNPs (red) and absorption spectra of PCL (black). **c** Protein release from Cre-UCNPs (50 μg/mL in PBS) was quantified by SDS-PAGE after activation by a NIR laser (785 mW/cm$^2$) for a specific time. Results are expressed as Mean ± SEM ($n = 3$ technical replicates). **d** Cellular uptake. Quantification of yttrium (Y) by ICP-MS in fibroblasts exposed to different concentrations of Cre-UCNPs for a specific time, washed, harvested and finally freeze-dried. The delivered dose was normalized per cell number. Results are expressed as Mean ± SEM ($n = 3$ independent experiments, being the value of each independent experiment the average of three technical replicates). **e** Cellular uptake mechanism. Uptake of Cre-UCNPs in fibroblasts after silencing key regulators of clathrin-mediated endocytosis (*CLTC* and *LDLR*), caveolin-mediated endocytosis (*CAV1*), GEEC-CCLIC pathways (*CDC42*), macropinocytosis (*RAC1* and *CTBP1*) and scavenger receptors (*SCARA3*) with siRNAs. Results were expressed as Mean ± SEM ($n = 3$ independent experiments) and analyzed by one-way ANOVA followed by Tukey's multiple comparisons test: (*), $P = 0.0229$; (**), $P = 0.0076$. **f** Intracellular trafficking. Co-localization of Cre-UCNPs (stained with anti-Cre antibody) with intracellular vesicles labeled for LAMP1 after 5 min exposure. Results are expressed as Mean ± SEM ($n = 17$–18 images obtained from 3 technical replicates in 3 independent runs) and analyzed by a two-tailed t-test: (*), $P = 0.0285$. **g** Endolysosomal disruption and formation of galectin-9 foci. Representative confocal microscopy images of fibroblasts 24 h after incubation with Cre-UCNP (with or without immobilized HCQ on their surface) or soluble HCQ (26 μg/mL; 60 μM) for 16 h. Scale bars = 20 μm. **h** Mean area of galectin-9 foci larger than 2 μm$^2$ and **i** total number of galectin-9 foci per cell. Results are expressed as Mean ± SEM ($n = 2$ independent experiments, each experiment with 2 technical replicates; 5-8 confocal images were acquired for each replicate). Statistical analysis was performed by one-way ANOVA followed by Tukey's multiple comparisons test: in **h**, (**), $P = 0.0084$; (***), $P = 0.0001$; in **i**, (**), $P = 0.0028$; (***), $P = 0.0008$.

endolysosomal membrane compartment[25]. To determine whether Cre-UCNPs with HCQ induced endolysosomal compartment disruption, we monitored galectin-9 by immunofluorescence, which is a very sensitive sensor of membrane damage and has been used to demonstrate endosomal escape of therapeutic molecules, including chloroquine[25]. Indeed, the cells exposed to Cre-UCNPs with HCQ showed an increased number of galectin-9 foci compared to those treated with Cre-UCNPs without HCQ (Fig. 2g–i). Moreover, the large foci size found in the cells treated

with Cre-UCNPs with HCQ further suggested the disruption of the endolysosomal compartment. These results also aligned with the results obtained by TEM analyses (Supplementary Fig. 7f, g). In contrast to Cre-UCNPs without HCQ, Cre-UCNPs with HCQ showed low colocalization within the endolysosomal compartment 4 h after transfection. Overall, our results indicated that Cre-UCNPs were taken up by cells mostly via clathrin-mediated endocytosis and that the NPs rapidly escaped the endolysosomal compartment due to the presence of HCQ.

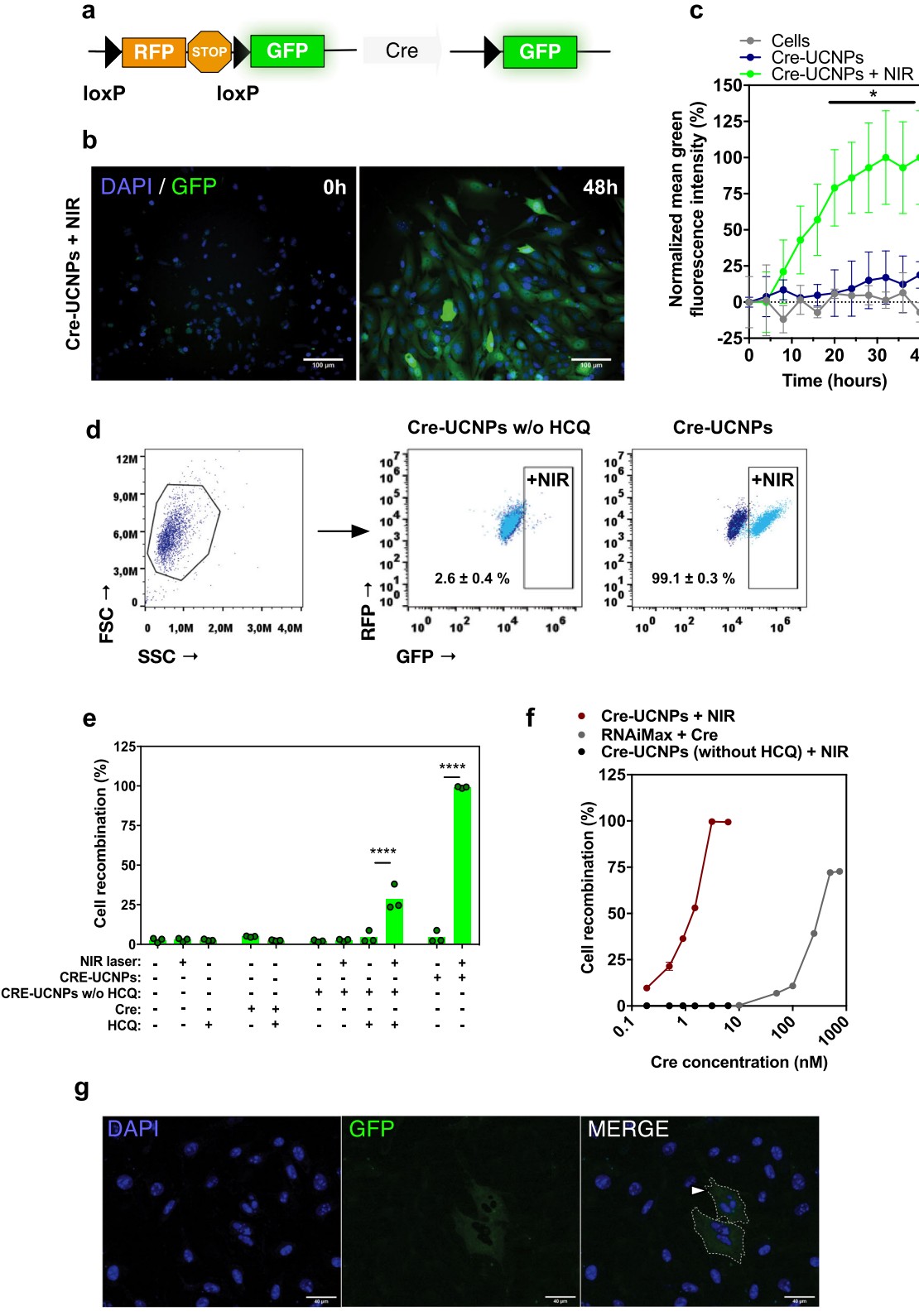

**Remote activation of Cre-UCNPs**. To evaluate the intracellular delivery of Cre recombinase by Cre-UCNPs, fibroblast Cre reporter cells containing a LoxP-RFP-Stop-LoxP-GFP sequence cassette (Fig. 3a) were exposed to the NPs for 4 h, washed to remove the noninternalized Cre-UCNPs and then activated by an NIR laser for 3 cycles of 5 min exposure. The recombination process was monitored by high-content microscopy (Fig. 3b, c and Supplementary Fig. 9a) and flow cytometry (Fig. 3d, e). In both analyses, the cells showed approximately 100% recombination after 30–40 h, even when cell transfection was performed in medium supplemented with 10% fetal bovine serum (Supplementary Fig. 9b). Importantly, when Cre-UCNPs without HCQ were evaluated, the recombination efficiency was found to be largely absent even when the NPs had been internalized by the cells (Fig. 3e and Supplementary Fig. 4a). In our experiments, 2.5 nM of Cre (i.e., 10 µg/mL Cre-UCNP) was sufficient to

**Fig. 3 In vitro activation of Cre conjugated into UCNPs. a** Reporter system for Cre activity. **b** Representative high-content microscopy images of the cell recombination process at time zero and 48 h after transfection, obtained from 3 independent experiments. Fibroblast reporter cells were transfected with Cre-UCNPs (50 μg/mL) for 4 h, washed, and activated by a NIR laser (980 nm; 785 mW/cm$^2$; 15 min). Scale bar = 100 μm. **c** Quantification of cell recombination. GFP fluorescence intensity was quantified and normalized to the background. Results are Mean ± SEM ($n$ = 3 independent experiments). Statistical analysis was performed by one-way ANOVA followed by Dunnett's multiple comparisons: (*), $P$ = 0.041. **d** Cell recombination monitored by flow cytometry. Representative scatter plots of cells treated with Cre-UCNPs at 40 h post-transfection. Fibroblast reporter cells were transfected with Cre-UCNPs without (50 μg/mL) or with HCQ (50 μg/mL; named as Cre-UCNPs) for 4 h, washed, and activated by a NIR laser. Percentage of GFP$^+$ cells was calculated based in the gates defined for untreated cells (below 3%; dark blue). **e** Quantification of cell recombination by flow cytometry. Results are Mean ± SEM (3 independent experiments; each experiment with 1–3 technical replicates). In Cre-UCNPs (50 μg/mL) the concentrations of immobilized Cre and HCQ were 12.5 nM (0.48 μg/mL) and 0.22 μM, respectively. Therefore, as a control, cells were treated with a solution containing the same concentrations of Cre and HCQ. In the experimental group of HCQ alone, the concentration was 100 μM. Cell recombination efficiency was calculated based in the percentage of GFP$^+$ cells. Statistical analysis was performed by one-way ANOVA test followed by a Bonferroni's multiple comparisons test: (****), $P$ < 0.0001. **f** Cell recombination efficiency by flow cytometry after cell transfection with Cre-UCNPs (with and without HCQ) and Cre using a commercial transfection agent (lipofectamine RNAiMAX). Results are Mean ± SEM ($n$ = 3 independent experiments). **g** Spatial control in cell recombination. Fibroblast reporter cells were transfected with Cre-UCNPs (50 μg/mL) for 4 h, washed and a single cell was activated in a multiphoton microscope by the NIR laser (marked by an arrowhead). Representative image of recombination at 48 h post-transfection from 3 independent experiments. Scale bar = 40 μm.

achieve recombination in 50% of the cells, which is a concentration approximately 125-fold lower than that used by the commercial transfection agent RNAiMAX, to achieve similar recombination efficiency (Fig. 3f). Our results compare favorably with other protein transfection agents, ranging from 4- to 100-fold enhanced activity compared to Cre enzyme engineered with a polyanionic protein[26], Cre complexed with cationic lipids[27], and Cre delivered by small transduction compounds after hypertonic shock[28] or cell-penetrating peptide TAT dimers[29]. Importantly, NIR activation was necessary to enable gene editing from intracellular Cre-UCNPs (Fig. 3e), which could be triggered up to 8 h after transfection, albeit with a reduction in their activity (Supplementary Fig. 9b). Furthermore, this nanoformulation enabled gene editing with single-cell resolution when multiphoton microscopy was used for NIR activation (Fig. 3g).

To evaluate NIR laser penetration, an initial in vitro assay was performed using murine tissue varying in thickness and composition placed above culture dishes containing a fibroblast Cre reporter line (Supplementary Fig. 10). We determined that a 980 nm laser at 780 mW/cm$^2$ power delivered through the whole mouse brain (thickness: ~6 mm) led to the generation of 1.4 mW/cm$^2$ of blue light (Supplementary Fig. 1d), which was sufficient to activate up to 50% of the Cre-UCNPs located within the Cre reporter cells (Supplementary Fig. 10). Next, an in vivo experiment was performed to evaluate NIR laser light penetration through live tissue. In this case, fibroblast Cre reporter cells (RFP-labeled without recombination) were preincubated with Cre-UCNPs and injected intramuscularly into the mouse hindlimb (approximately 4 mm deep). We chose intramuscular injection since we could readily trace and characterize the implanted cells. Immediately after the injection, the Cre-UCNPs were activated with NIR light, the muscles were harvested after 60 h, the number of GFP-positive cells was determined, and the GFP intensity was quantified (Supplementary Fig. 11). Implanted cells activated with an NIR laser had a higher number of GFP-positive cells than cells without activation. Collectively, our results indicated that we had designed a formulation that can be activated through biological tissues up to a few millimeters deep after in vivo administration.

**Gene editing of SVZ cells mediated by Cre-UCNPs.** Brain delivery of gene-editing protein formulations offers the potential to interrogate brain circuits and potentiate in vivo regenerative processes. Viral delivery of Cre-expressing constructs by stereotaxic administration into the brain or by crossing a cre-driver mouse line with a cre-dependent reporter mouse line has been performed; however, these may suffer limitations in terms of spatial resolution and biocompatibility issues (see below).

Alternatively, in vivo gene editing of the inner ear and brain cells has been demonstrated by the delivery of Cre recombinase fused with a polyanionic protein and complexed with cationic lipids[26,30], or by encapsulation in bioreducible lipid NPs[27]. However, these formulations have limited transfection efficiency (see above); they require the fusion of Cre with a negatively charged protein and they do not enable spatial control over gene editing. To demonstrate the potential for gene editing of the Cre-UCNPs in brain cells, we chose to target SVZ cells since these cells are an important in the adult mammalian neurogenic niche[31,32]. Cells were isolated from Rosa26-YFP neonatal transgenic mice[33], which contain a LoxP-Stop-LoxP-YFP sequence cassette inserted into the Gt(ROSA)26Sor locus (Fig. 4a). SVZ cells were cultured in the presence of Cre-UCNPs for 4 h, washed to remove the noninternalized NPs, exposed or not exposed to an NIR laser, and cultured for 4 additional days in fresh culture medium to allow cell recombination (Fig. 4b). The transfected cells showed similar nuclear condensation at all the concentrations tested, indicating low cytotoxicity; however, higher concentrations of Cre-UCNPs affected SVZ cell proliferation (Supplementary Figs. 12a–b). Cellular uptake of the Cre-UCNPs was demonstrated by ICP-MS analyses as well as Cre immuno-labeling (Supplementary Figs. 12c–e). Cell recombination was evaluated by the quantification of YFP-positive cells, using a high-content microscope (Fig. 4c). Notably, Cre-UCNPs mediated gene editing with higher efficiency than both soluble Cre (even with HCQ) and adeno-associated transfection with Cre (AAV5-Cre; used in this study as a control). These results are in line with previous reports showing limited AAV5 efficiency in mediating gene delivery into some stem cells[34,35]. Next, we investigated the tropism of our formulation by characterizing the phenotype of the resulting YFP-positive cells (Fig. 4d, e). YFP-positive cells were found to be equally distributed within more immature cells (Nestin$^+$ and Nestin$^+$GFAP$^+$ cells) and neuronal differentiated cells (NeuN$^+$ cells), whereas fewer GFAP-positive astrocytes were found.

To further demonstrate the potential of Cre-UCNPs to control endogenous SVZ cells, NPs were administered to the SVZ region of Rosa26-YFP mice by stereotaxic injection. One hour after administration, a subset of animals was sacrificed to assess the intracellular accumulation of Cre-UCNPs by TEM analyses (Supplementary Fig. 13a–b) and some of the animals were stimulated with or without exterior NIR light (Fig. 4f). One month following activation, the YFP signal was confirmed by immunostaining (Fig. 4g), and the fluorescence intensity and the percentage of YFP-positive cells were quantified in the region of interest (Fig. 4h, i). Our results showed Cre-UCNP-mediated

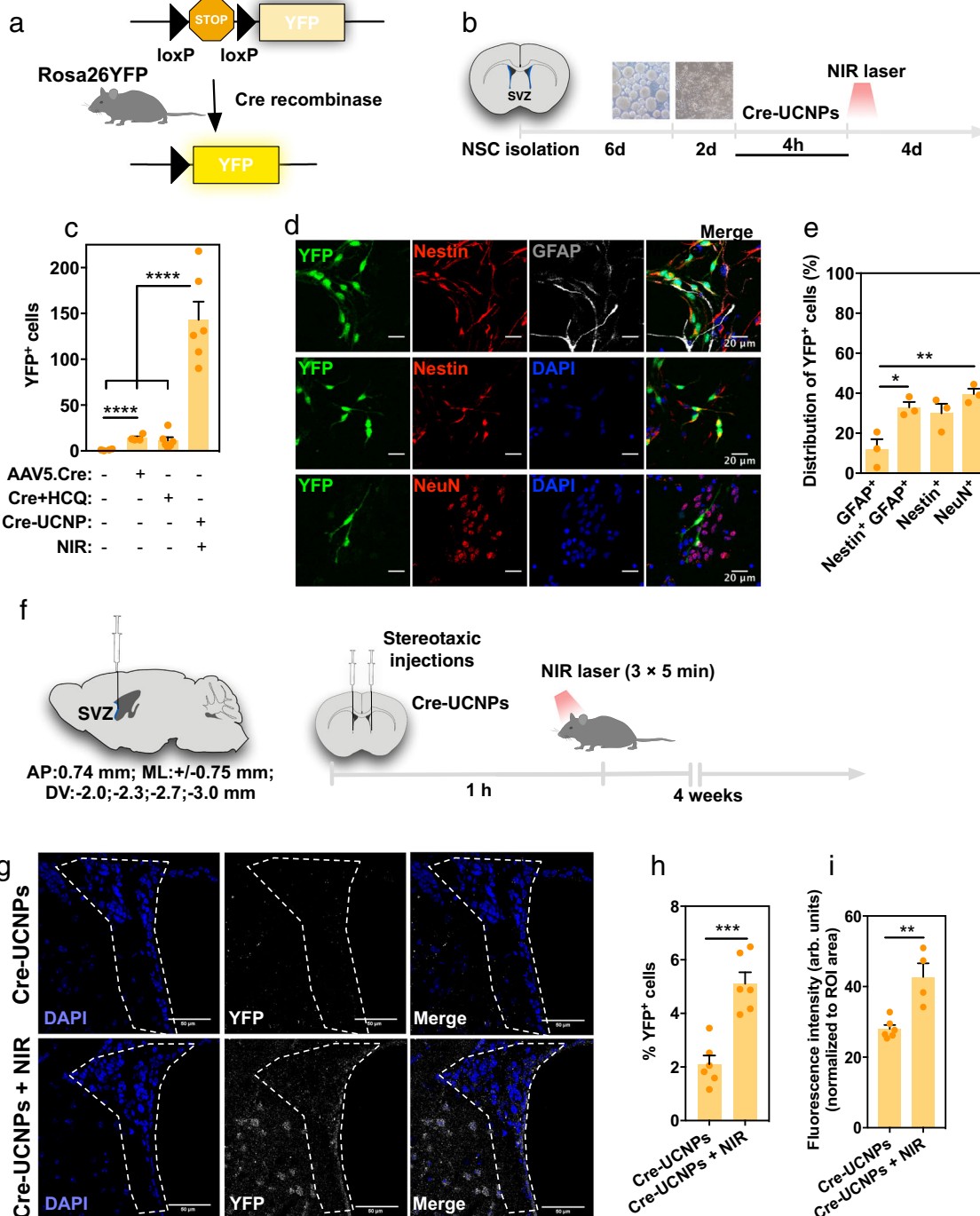

**Fig. 4 Gene editing at SVZ region mediated by Cre-UCNPs. a** In vivo reporter system for Cre activity. The administration of Cre-UCNPs in the SVZ region of R26YFP mice, followed by the NIR laser exposure leads to the excision of a stop codon and consequent expression of YFP fluorescence reporter protein. **b** Schematic representation of the in vitro protocol to study the gene edition of SVZ cells by Cre-UCNPs. **c** Quantification of YFP-positive cells in SVZ cells transfected with: (i) Cre-UCNPs (200 μg/mL) followed by NIR laser activation, (ii) AAV or with (iii) soluble Cre in the presence of HCQ (concentrations equivalent to the cargo of 200 μg/mL Cre-UCNPs). Results are Mean ± SEM (2-3 independent experiments, each experiment with 2 technical replicates). **d** Colocalization of YFP with markers of SVZ cell sub-populations. YFP positive cells (green) were immunolabelled for Nestin (red), GFAP (gray) and NeuN (red). Representative images from 3 independent experiments. Scale bar = 20 μm. **e** Distribution of YFP-positive cells within SVZ cell sub-populations. Results are Mean ± SEM (n = 3 independent experiments, each one is average of two technical replicates). In (**c**) and (**e**), statistical analyses were performed by one-way ANOVA test followed by a Tukey´s multiple comparison test: (*), $P = 0.0244$; (**), $P = 0.0052$; (****), $P < 0.0001$. **f** Schematic representation of the animal protocol used to monitor in vivo gene editing in the SVZ region, 4 weeks after stereotaxic injection of Cre-UCNPs.
**g** Representative confocal microscopy images of the SVZ region, from n = 2–3 animals. Cells labelled in yellow (stained by an anti-YFP antibody) were edited by Cre recombinase. Scale bar = 50 μm. **h** Percentage of anti-YFP stained positive cells. **i** Quantification of anti-YFP fluorescence intensity in SVZ region tissues. Results are Mean ± SEM (n = 2–3 animals per condition, 2 injections per animal, 4 photos per injection; intensity normalized to ROI). Data in **h** and **i** were analyzed by an unpaired two-tailed t test: (**), $P = 0.0027$ (***), $P = 0.0002$.

gene editing in the SVZ region with various SVZ cell subpopulations, indicating recombination (Supplementary Fig. 13c). Altogether, our results showed successful gene editing of SVZ cells.

**Coupling gene edition, mediated by Cre-UCNPs, with optogenetics.** Optogenetics is based on optical stimulation and genetics to achieve cell-type-specific modulation of electrical and biochemical neural activity, often with high temporal resolution[36,37]. To achieve this effect, light-activated channels, such as microbial opsins, are artificially introduced into the genome of the cells. The most widely used method to incorporate opsins into the regions of interest is using AAV vectors for transport and delivery. However, this strategy has limitations since the virus often leaks from the region of interest to other locations, leading to misinterpretation of the biological activity[38], and gene delivery is relatively low in some brain cells, such as NSCs[34] or microglia[39]. Cre-UCNPs might be used to overcome these limitations since NPs offer greater spatial resolution when injected in vivo and have broader applicability since they are taken up by SVZ cells (see previous section) as well as microglia[16]. In addition, a recent demonstration showing that UCNPs can act as optogenetic actuators of transcranial NIR light to stimulate deep brain neurons[16] suggested new opportunities for the combination of gene editing with optogenetics. Therefore, to evaluate in vitro cellular activity with optogenetics after gene editing by Cre-UCNPs, we transfected primary cortical neurons with AAV5 carrying Cre-dependent ChR2, followed by a second transfection with Cre-UCNPs and NIR laser stimulation to release Cre and initiate the recombination process (Fig. 5a, b). The concentration of Cre-UCNPs (100 µg/mL) was selected based on a compromise between cellular uptake capacity and NP cytotoxicity (Supplementary Figs. 14a–c). Three days after irradiation, 25.6% of the cells were positive for YFP and ChR2 (Fig. 5c, d, Supplementary Fig. 14d and 15). This percentage is close to the 31.7% observed for cells transfected with AAV encoding YFP (not requiring Cre-mediated recombination to express YFP). The activation of ChR2 was demonstrated by whole-cell patch-clamp experiments (Fig. 5e, f). Approximately 300 pA of electric current was generated in cortical neurons after blue light activation (power density: 4.2 mW/cm$^2$) of ChR2 (Fig. 5e). The electrophysiological traces presented a characteristic initial peak followed by a steady-state current. As expected, the amplitude of the generated currents was correlated with the blue light power (Fig. 5f). However, while blue-upconverted light generated after NIR irradiation of UCNPs can potentially be used for inducing ChR2 activity[16], we were unable to specifically dissect the activation of ChR2 due to the depolarization of other direct NIR-mediated membrane channels (independent of ChR2 expression) in cortical neurons (Supplementary Fig. 16).

To evaluate in vivo brain cell activity with optogenetics after gene editing, Cre-UCNPs were injected into the VTA; this region is located deep in the brain (approximately 4.0 mm from the bregma[16]) and presents well-characterized anatomy and function, containing dopaminergic neurons that have been implicated in reward mechanisms[40]. To express ChR2, AAV5.DIO.ChR2.YFP particles (Supplementary Table 1) and Cre-UCNPs were injected simultaneously into the VTA via stereotaxic administration in the VTA (Fig. 6a). The concentration of UCNPs in the brain decreased over time, up to 7 days post injection, as confirmed by ICP-MS analyses (Fig. 6b). No significant microglia activation was observed in the VTA region on day 3 post administration (Supplementary Fig. 17).

Cre-UCNPs successfully released Cre after NIR activation and thus induced ChR2 expression in the VTA dopaminergic neurons,

as confirmed by the expression of YFP and tyrosine hydroxylase (TH) in the injected region (Fig. 6c, d). Importantly, ChR2 expression was markedly restricted to the VTA region after gene editing with the Cre-UCNPs (Supplementary Fig. 18), in contrast to the pattern observed with AAVs. To evaluate the impact of the selective expression of ChR2 in the VTA region, we monitored mouse behavior after blue light-mediated depolarizations of the VTA using a conditioned place preference (CPP) behavioral test. This conditioning test is widely used to determine the effects that specific objects or experiences exert on reward or aversion behaviors in laboratory animals[37,41]. In our setup, a standard home cage was modified with two areas that might elicit contrasting visual (black vs. white) and tactile (sandpaper vs. grid) reactions (Fig. 6a). The first part of the trial (Pretest) consisted of 3 days of habituation to determine endogenous preferences for a zone in the arena. Next, on the fourth day (Conditioning), we used a closed-loop stimulation interface to optogenetically activate the VTA whenever the animal crossed into the unpreferred zone. A final probe trial (Test) was performed on the fifth day to assess any preference changes. Our results showed that the animals exhibited changed preferences after optical stimulation compared to that exhibited by the control animals, confirming an optogenetic induced reward-based conditioning (Fig. 6e). We further validated our hypothesis that VTA dopaminergic neurons were activated following ChR2 stimulation by evaluating c-Fos expression[42] (Fig. 6f, g). After Cre-mediated recombination, ChR2-expressing mice exhibited more c-Fos-positive cells in the VTA region than control mice (with no ChR2 expression).

Despite the demonstration of in vivo gene editing and optogenetic actuation in freely moving animals, one of the main limitations of the presented strategy is the requirement of an invasive administration route. Therefore, we investigated the potential of using Cre-UCNPs to perform targeted gene editing through a non-invasive route, specifically intranasal instillation (Supplementary Fig. 19). For this purpose, R26tdTomato mice were subjected to 3 intranasal administrations of Cre-UCNPs on 3 different days (i.e. days 1, 2 and 3). Each day, drops of NP suspensions were administered gradually into both nasal cavities (Supplementary Fig. 19a). The animals were divided into two groups: the Cre-UCNP group received NPs (120 µg per administration), and the control group received only a saline solution.

First, we investigated the distribution of the Cre-UCNPs 4 h after a single intranasal administration by ICP-MS analyses (Supplementary Fig. 19b). Previous studies have demonstrated that nanomaterials administered through the nasal route can reach the brain[43,44]. Nose-to-brain transport has been reported for several nanomaterials, including exosomes[43] and iron oxide nanoparticles[44], among others, and can be primarily described by two mechanisms: intracellular (via transcytosis) and extracellular. The extracellular pathway is the faster route and includes the passage of the NPs by passive transport through tight junctions into the brain parenchyma[45]. The intracellular pathway includes endocytosis of the NPs into olfactory and trigeminal nerve branches followed by axonal transport into brain. Our results show that most of the NPs were in the nasal cavity, followed by the olfactory bulb (Supplementary Fig. 19b). Because Cre-UCNPs reached the cortex and hippocampus within a few hours, it is possible that the NPs reached these parts of the brain by extracellular transport.

Next, we tested whether Cre-UCNPs induced gene editing with unilateral specificity by exposing only one of the brain hemispheres to NIR light (the other was covered to prevent laser activation; 980 nm; 425 mW/cm$^2$; 3 cycles of 5 min each) at 1 h after administration (Supplementary Fig. 19c, d). Our results showed recombined cells in the irradiated brain hemisphere,

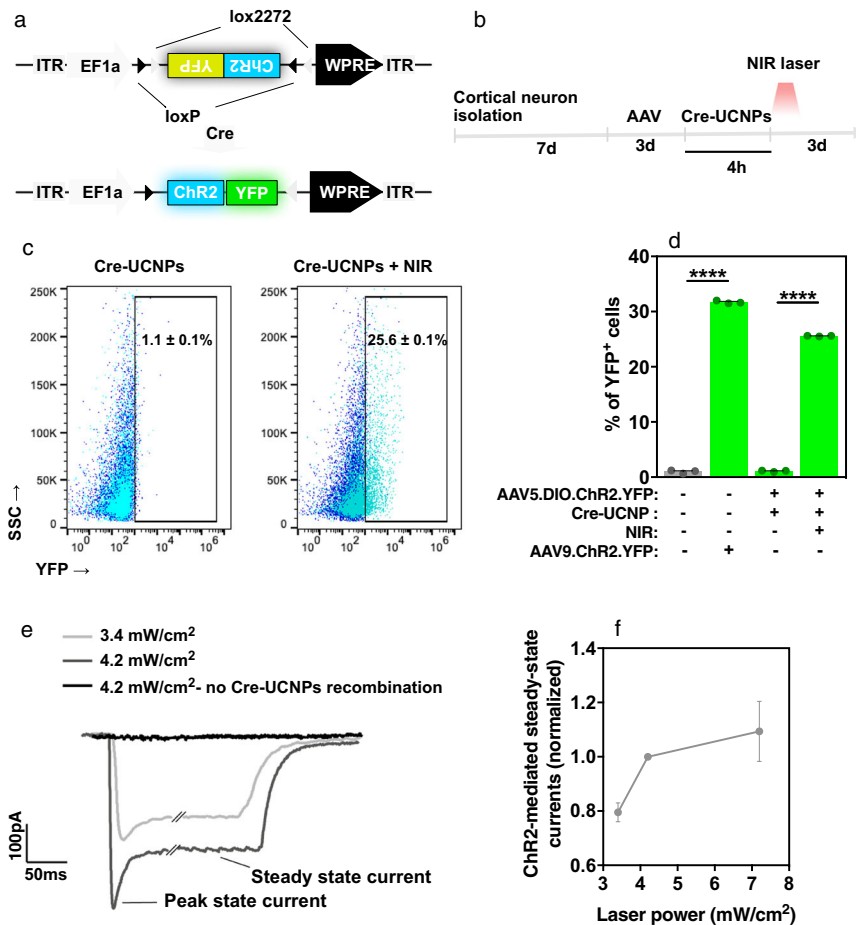

**Fig. 5 In vitro gene editing of cortical neurons by Cre-UCNPs followed by cellular activity measurements with optogenetics. a** AAV.dflox.ChR2-YFP cassette used for Cre-mediated expression of ChR2 in cortical neurons. Cre-UCNPs delivery to cells followed by NIR laser activation leads to the inversion of the double-floxed inverted open reading frame and results in the expression of ChR2-YFP fusion protein. **b** Schematic representation of the experimental setup with cortical neurons. After isolation, cells were cultured for 7 days before treatments. Viral particles (AAV5.dflox.ChR2-YFP) were added to the medium for 3 days. Cells were then exposed to Cre-UCNPs (100 μg/mL) for 4 h, activated by NIR laser (980 nm; 785 mW/cm$^2$; 3 cycles of 5 min irradiation) and allowed to recombine for 3 days. Neurons were activated by a blue light for electrophysiology measurements. **c** Flow cytometry analyses of ChR2-YFP expression in neuronal cultures. Cells were transfected with Cre-UCNPs and activated or not with a NIR laser. Positive cells were gated based in the basal expression of ChR2-YFP in nontreated cortical neurons (dark blue). **d** Quantification of recombination by flow cytometry. Results are Mean ± SEM ($n = 3$). Statistical analyses were performed by one-way ANOVA followed by Bonferroni's multiple comparisons test: (****), $P < 0.0001$. **e** Representative traces of ChR2-mediated currents, showing dependency on the intensity of blue light. Scale bars = 100 pA, 50 ms. Darker trace shows the dependence of Cre-UCNP recombination on the generation of photocurrents. **f** Voltage-clamp currents generated from different blue-light stimulus. Results are Mean ± SEM ($n = 3$ independent experiments).

particularly in the olfactory bulb and cortex (Supplementary Fig. 19c). The area of tdTomato-positive cells in the olfactory bulb of the irradiated hemisphere was greater than that in the nonirradiated hemisphere (Supplementary Fig. 19d). Overall, our results demonstrated the utility of Cre-UCNPs for on-demand gene editing and the possibility to manipulate edited cellular circuits with optogenetic activation.

## Discussion

We have developed a delivery system for gene-editing enzymes that is capable of efficient endolysosomal escape and on-demand release into deep tissues with high spatial resolution upon exposure to transcranial NIR light. This formulation enabled transient expression of Cre recombinase with 125-fold higher efficiency than commercially available lipofectamine and did not require further protein engineering to facilitate cellular uptake[26,27]. We have demonstrated the utility of the formulation in vivo with the achievement of gene editing in neurogenic niches of the mouse brain, as well as the activation of mechanisms of

reward and reinforcement in the VTA using NIR activation to genetically unlock an optogenetic actuator (ChR2), followed by its activation by blue light. The formulation reported here might be used in the treatment of reward-pathway dysfunctions that are highly relevant in psychiatric (e.g., substance-use, affective, eating and obsessive-compulsive disorders) and neurodevelopmental disorders (e.g. schizophrenia, attention-deficit/hyperactivity disorder and autism spectrum disorders), and genetic syndromes (i.e, fragile X syndrome, Prader-Willi syndrome, Angelman syndrome, and Rett syndrome)[46–48]. In addition, our formulation might be of particular relevance for gene editing in the context of the eye. For example, optogenetic modulation of the retina with the AAV-mediated expression of channelrhodopsin and subsequent visual stimulation for the treatment of retinitis pigmentosa is currently being assessed in a clinical trial (NCT03326336). Moreover, considering the large number of Cre/lox mouse lines available, it is anticipated that the formulation developed in this work might be useful for researchers to map specific neuronal circuits in vivo, as well as to image and track specific cell

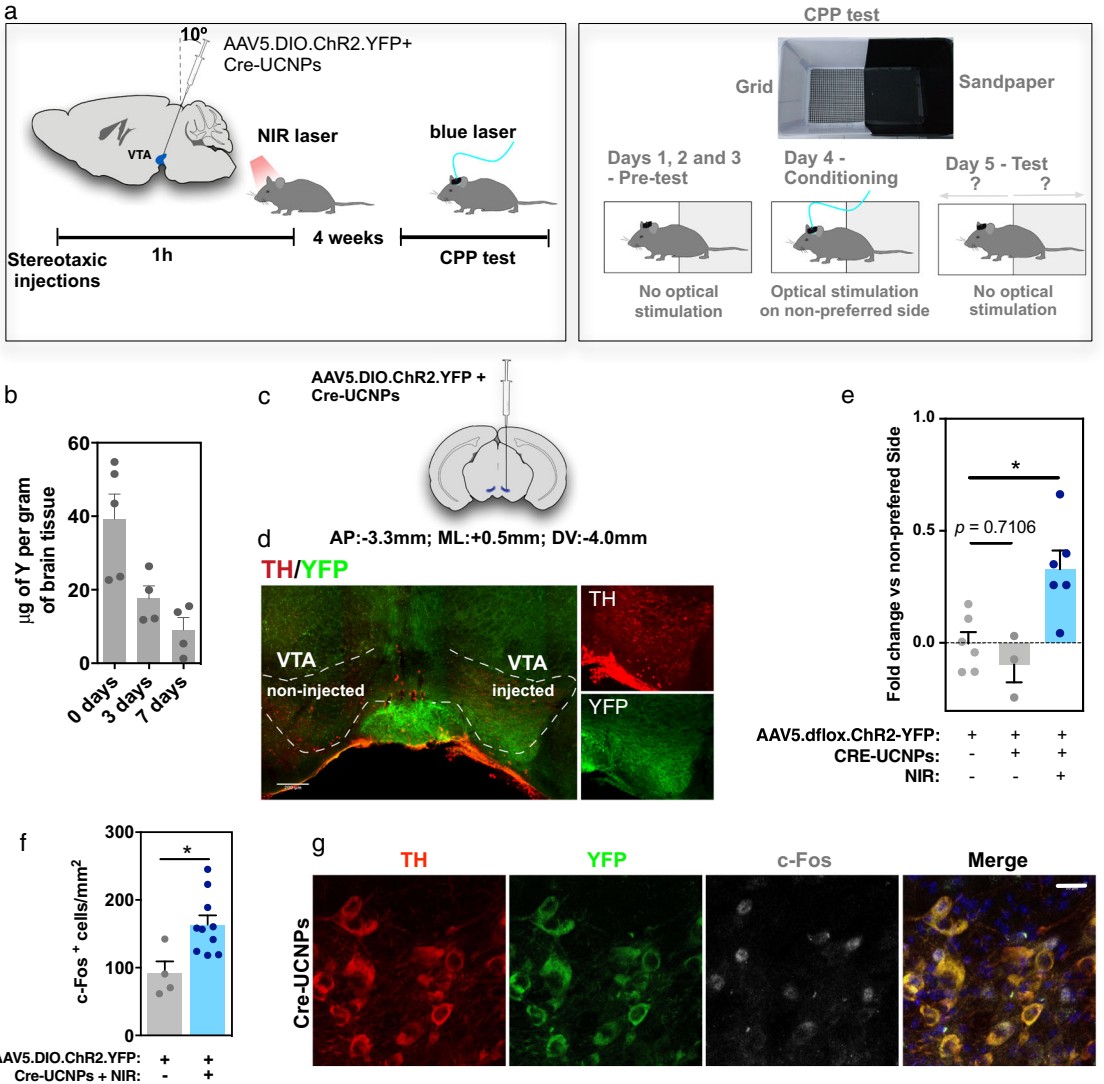

**Fig. 6 In vivo gene editing with Cre-UCNPs followed by functional analyses of the edited cells by optogenetics. a** Schematic representation of the in vivo setup. AAV5.dflox.ChR2-YFP was injected into the VTA together with Cre-UCNPs (30 μg). Following stereotaxic injection, the target site was irradiated by a NIR laser (980 nm, 425 mW/cm² for 3 cycles of 5 min) to induce the release of Cre from the NPs. After 4 weeks, animals were subjected to a CPP test, where the animal's preference for one of two regions with contrasting visual and tactile cues in a cage was monitored for 3 days using an automated mouse tracking system. On day 4, whenever the mouse crossed into the non-preferred region of the cage, the VTA was activated with blue light; on the final test day, the animals were re-introduced into the cage to assess changes in preferences. **b** Quantification of Cre-UCNPs accumulated in the brain of mice was performed in dried tissue by ICP-MS at days 0, 3, and 7 post stereotaxic administration. Results are Mean ± SEM (n = 4–5 animals). **c** Local of injection of Cre-UCNPs. **d** Expression of ChR2-YFP (green) in the TH-positive region (red) of the VTA, as evaluated by immunofluorescence. Of note the differences in fluorescence between left and right VTA (*n* = 6 animals). **e** CPP test to evaluate behavioral changes in preference (fold change in the time spend on the non-preferred chamber before and after blue light stimulation). Results are Mean ± SEM, obtained from n = 3 (CRE-UCNPs), or 6 (Negative control and CRE-UCNP + NIR) animals. Statistical analysis was performed by one-way ANOVA followed by Tukey's multiple comparisons test: (*), *P* = 0.0115.
**f** Expression of c-Fos positive nuclei in the VTA, following stimulation with blue light. Results are expressed as Mean ± SEM (2–5 animals per condition; for each animal, c-Fos was quantified in 2–3 regions in VTA slices, 4-13 fluorescence images per slice) and analyzed by unpaired two-tailed t test: (*), *P* = 0.024. **g** Representative confocal images of a mouse treated with Cre-UCNPs showing the expression of c-Fos (gray) and ChR2-YFP (green) in the TH positive region (red) of the VTA, from n = 5 animals. Scale bar = 20 μm.

populations[49]. Finally, UCNPs accumulated in the brain may be used as optogenetic actuators of transcranial NIR light to stimulate deep brain neurons, as has been recently demonstrated[16,50].

The efficiency of our formulation relied on the incorporation of low amounts of HCQ chemically conjugated to the UCNPs (below 1 μM) that induce endolysosomal escape. Previously, it has been shown that soluble chloroquine is one of the most effective agents to disrupt the endolysosomal compartment[25].

The molecule induced the disruption of endolysosomes that were positive for late endosomal markers including CD62/Rab7 (~30%) and Rab9 (~23%), and the lysosomal marker LAMP1 (~45%)[25]. Our results indicated that, in the absence of light activation, Cre-UCNPs with HCQ, but not Cre-UCNPs without HCQ, efficiently escaped the endosomal compartment at levels similar to those of soluble HCQ. To demonstrate this escape, we have used galectin-9 protein as a reporter, since it has been shown to respond rapidly and with high sensitivity to endosomal

membrane disruption[25]. This protein is recruited to damaged endosomes during lipid NP or transfection agent-mediated endosomal escape[25]. Our results showed that very small amounts of immobilized HCQ in NPs can be as effective as high concentrations of soluble HCQ (60-fold higher). It is possible that the immobilized HCQ molecules acted synergistically in the endosomal membrane, maximizing its effect. Importantly, we anticipate that the strategy described here to facilitate endosomal escape based on immobilized HCQ can be easily incorporated into other delivery systems for protein or noncoding RNA delivery.

The results reported here demonstrate our capacity to control the activity of genes with precise spatial control, using a Cre-based NP formulation that can be externally triggered using NIR light. First, the capacity of Cre-UCNPs to induce in vitro cell recombination, with spatial resolution, was demonstrated at single-cell level (Fig. 3g). To the best of our knowledge, this level of control has not been demonstrated in the past using NIR light-triggerable NP formulations[12,51,52]. Second, we showed the capacity of Cre-UCNPs to induce in vivo cell recombination, with spatial resolution at the hemisphere level (Supplementary Fig. 19). In this case, the spatial control in gene editing was controlled by the spot of the light and not defined by the point of injection (as demonstrated in the SVZ and VTA studies). In the near future, it might be possible to align the laser with a stereotaxic equipment to activate a particular region of the brain. For applications requiring deeper penetration of the light (more than 4 mm), a craniotomy (a clinical procedure that is used in humans to implant drug delivery systems in the brain[53]) may be required. Third, our results further showed that the spatial control obtained by Cre-UCNPs in the VTA region was higher than that obtained with an AAV5 virus (Supplementary Fig. 18). After injection into the target area of the VTA, the AAV virus spread throughout the tissue with ChR2 expression appearing at distances of 3.5 mm from the injection site, while with the Cre-UCNPs, the expression of ChR2 was 1.5 mm from the injection site.

Our results demonstrated that the activity of the edited cells can be monitored by optogenetics using blue light as a trigger. Although not tested in the current study, the modulation of brain cell activity may be performed by chemogenetics[54] rather than optogenetics. In chemogenetic applications, cells are transfected with DREADD receptors that are activated by synthetic ligands that do not require invasive administration. In addition, it might be possible to incorporate transcription factor genes into the Cre-LoxP construct, in addition to ChR2, to investigate the impact of these factors in brain cells at the functional level. Importantly, using behavioral tests, we showed that the level of gene editing in the brain was sufficient for a functional output.

Our formulation may represent a combined alternative with reduced safety problems in the rhodopsin delivery strategy and efficient NIR-mediated stimulation. A widely used approach to study gene function in vivo is Cre/loxP recombination through viral delivery of Cre-expressing constructs. However, even though the widespread use of this technique is an indication of its usefulness, several studies have reported nonspecific and potentially noxious effects of Cre recombinase in mice in various organs, including the brain[55,56]. In addition, a recent study demonstrated that local AAV-mediated Cre expression in the substantia nigra of wild-type mice induced a massive decrease in neuronal populations, greatly impacting the nigrostriatal dopaminergic pathway, culminating in an anatomical and functional phenotype resembling models of dopaminergic degeneration[57]. Another method to achieve Cre/loxP recombination is through Cre-driver mouse lines, where optogenetic tools, delivered via Cre-inducible viral constructs, are expressed only in defined neurons expressing Cre recombinase. However, the use of these mouse lines has

drawbacks. For example, nonspecific expression of Cre recombinase may occur outside the target cell or brain region due to promiscuous activity of the driving promoter[58,59]. In addition, Cre-driver lines can present inherent physiological problems, such as altered metabolic phenotypes, impacting the body length and body weight of the mice[60]. These findings underline the necessity for in-depth scrutiny of Cre toxicity and spatial localization in the brain, and further affirm our formulation as a possible tool to mitigate the side effects of existing techniques.

During the preparation of this manuscript, NIR-activatable NPs able to release Cre recombinase with spatiotemporal control were described[12]; however, the formulation showed low gene editing (<20% of cell recombination), and the in vivo activity of the formulation was not demonstrated. Other recent studies have also documented the optical regulation of gene-editing formulations based on NIR. For example, an upconversion-activated CRISPR-Cas9 NP has been reported for the editing of cancer cells transplanted into skin tissue[51]. However, the study did not: (i) show the spatial resolution of the technology in vitro or in vivo (to evaluate the functional impact of their formulation, the authors injected the NPs in the tumor site, irradiated the tissue, and measured the reduction in tumor size), (ii) demonstrate gene editing with spatial resolution using a noninvasive route for the administration of the NPs, and (iii) did not evaluate the cellular activity of the edited cells by optogenetics. Another study has reported a nanoparticle formulation for programmable genome editing in the second near-infrared (NIR-II) optical window[52]. The formulation was composed of a cationic polymer-coated gold nanorod and a Cas9 plasmid driven by the heat-inducible promoter HSP70. The authors have shown the capacity to regulate Cas9 activity by heat-induced gene expression. Although this system may show greater tissue-penetration than NPs that respond to the first near-infrared optical window (such as Cre-UCNPs), it suffered from some limitations such as: (i) it was based in a plasmid and not in a protein (in contrast to the Cre-UCNPs), (ii) the long-term effects of the intracellular fluctuation of the temperature on cell activity remain to be determined, and (iii) the transfection efficiency was only 3-fold higher than that of lipofectamine, while in Cre-UCNPs the transfection efficiency was more than 100-fold higher than that of a lipofectamine-based transfection agent (RNAiMax). Overall, the formulation reported here opens new avenues for in vivo spatial control of gene expression, which can be complemented with optogenetics to measure the activity of the edited cells.

Future studies should be directed to improving the accumulation of Cre-UCNPs in the brain after intranasal instillation to achieve high cell recombination. Additionally, while the Cre enzyme was used in this study as a proof of concept in this study for active protein delivery, we foresee that the designed NP formulation can be useful for the delivery of other gene editing proteins, such as CRISPR-Cas9[61], as well as transcription factors for cell reprogramming. In the case of Cas9, our formulation might offer specific advantages to correct mutations in SVZ cells since they have been recently associated with the development of glioblastoma[62].

## Methods

**Ethical statement**. All animal experiments were carried out in line with the ARRIVE guidelines, with prior ethical approval from the Portuguese DGAV, under project license 0421/000/000/2020, and from the ethics committee of the Center for Neuroscience and Cell Biology, University of Coimbra (Ref: ORBEA_262_2020/25052020), in accordance with Portuguese Decree-Law no. 113/2013 and the European Directive 2010/63/EU regarding animal use in research.

**Preparation of UCNPs: synthesis of the nanocrystal**. The UCNPs (NaYF$_4$:Tm$^{3+}$ 0.5 mol%, Yb$^{3+}$ 30 mol%) were synthesized according to a protocol published elsewhere[19] with minor modifications. The Yb$^{3+}$ acts as the sensitizer, while the

$Tm^{3+}$ acts as the activator ion. Tm ions were selected as an activator in order to enable a blue to UV-shifted emission, while Yb ions was selected because it has only two multiplets, which energy gap between the ground state and excited state offers efficient absorption in the range of $900 - 1000$ nm. For the synthesis of nanocrystals, thulium(III) acetate hydrate (Sigma-Aldrich, 1.7 mg, 0.015 mmol), ytterbium(III) acetate hydrate (Sigma-Aldrich, 38.1 mg, 0.9 mmol), and yttrium(III) acetate hydrate (Sigma-Aldrich, 55.5 mg, 2.1 mmol) were dissolved in a mixture of 1-octadecene (45 mL, Acros Organics) and oleic acid (Alfa Aesar, 18 mL). The resulting solution was slowly heated to 120 °C under vacuum with magnetic stirring for 30 min to remove residual acetic acid, water and oxygen during which time the flask was purged periodically with nitrogen gas. The resulting clear solution had a slight yellow colour indicating the formation of lanthanide oleate complexes. The temperature was then lowered to 50 °C and the reaction flask placed under a gentle flow of nitrogen gas. A solution of ammonium fluoride (12 mmol) and sodium hydroxide (7.5 mmol) dissolved in methanol (30 mL) was prepared via sonication. This methanol solution was added to the previous reaction flask and the resulting cloudy mixture was stirred for 30 min at 50 °C. The reaction temperature was then increased to 70 °C and the methanol distilled off from the reaction mixture. Subsequently, the reaction temperature was increased to 300 °C (in a period of 2-3 min) and maintained at this temperature for 90 min under the nitrogen gas flow. During this time, the reaction mixture became progressively clearer, having a yellowish-brown colour. The mixture was allowed to cool to room temperature and the UCNPs were precipitated by the addition of acetone and isolated via centrifugation (3000 rpm, 1157 g, 6 min). The resulting pellet was dispersed in a minimal amount of cyclohexane and precipitated with excess anhydrous ethanol. The UCNPs were isolated via centrifugation (3000 rpm, 1157 g, 6 min) and then dispersed in cyclohexane (30 mL). Next, a $NaYF_4$ shell was grown in the previous UCNPs. The dopant was yttrium within a sodium fluoride ($NaF_4$) matrix. The sodium fluoride matrix was chosen since it possesses a close lattice match to the dopant ions and has low phonon energy. The incorporation of the $NaYF_4$ shell is vital since it improves the intensity of the core UCNPs by minimizing the effect of the quenching due to interactions of dopant ions with the solvent. In brief, 71.8 mg (2.7 mmol) of yttrium(III) acetate hydrate was dissolved in a mixture of 1-octadecene (45 mL, Acros Organics) and oleic acid (18 mL). The temperature was increased to 120 °C for 30 min. Next, the temperature was decreased to 80 °C to introduce the $NaYF_4$:Tm, Yb seeds in cyclohexane. The temperature was increased to 110 °C to remove the cyclohexane solvent. The temperature was decreased to 50 °C for the addition of $NaF_4$ matrix (a solution of ammonium fluoride (10.5 mmol) and sodium hydroxide (6.6 mmol) dissolved in methanol (30 mL) was prepared via sonication) for 30 min. The temperature was increased to 70 °C to remove the methanol and then increased to 300 °C for 90 min. The nanocrystals were precipitated by the addition of acetone and isolated via centrifugation. The resulting pellet was dispersed in a minimal amount of cyclohexane and precipitated with excess anhydrous ethanol. The nanocrystals were isolated via centrifugation and then dispersed in cyclohexane (20 mL).

**Preparation of UCNPs: reaction of UCNPs with organosilanes containing terminal $N_3$ and $NH_2$ groups.** Initially, ligand exchange[63] was performed to replace the hydrophobic oleic acid on UCNP surface by citric acid to obtain core-shell UCNPs that can be dispersed in water. The core-shell UCNPs coated with oleic acid (580 mg) were dispersed in a mixture of 1,2-dichlorobenzene and N, N'-dimethylformamide (DMF) (80 mL, 1:1, v/v, Sigma-Aldrich). The mixture was stirred at 100°C for approximately 24 h after the addition of citric acid (0.5 g, Sigma-Aldrich). The hydrophilic particles were precipitated by the addition of ethyl ether. The UCNPs were washed by dispersion in acetone and centrifugation three times (4000 rpm, 2057 g, 6 min). Finally, the UCNPs were dispersed in water (30 mL). The suspension of core-shell UCNPs (30 mg, 200 μL) was added to methanol (15 mL), followed by the addition of a solution of APTES (6 μL) and $N_3$PTES (12 μL, Supplementary Fig. 20) in methanol (15 mL) and stirred for 12 h at room temperature. The NP suspension was then added to glycerol (30 mL) to prevent the aggregation of the NPs during the evaporation of methanol and water (by rotary evaporator respectively, at 40 and 80°C) and the dispersion was condensed in vacuum at $100 - 110$°C for 2 h. Flocculated APTES/$N_3$PTES-modified UCNPs were when washed with water/acetone (3:7, v/v, 100 mL, three times), being in each separated from the solvent by centrifugation (6000 rpm, 4629 g, 6 min). Finally, the modified UCNPs ($N_3$/$NH_2$-UCNPs) were suspended in water (1.5 mL).

**Preparation of UCNPs: modification of $N_3$/$NH_2$-UCNPs with hydroxychloroquine.** Hydroxychloroquine sulfate (27 mg, 62.5 μmol), was suspended in a 1,1"-carbonyldiimidazole solution (CDI, 67 mg/mL in THF; 750 μL) and the slurry taken to 40°C under argon atmosphere and gentle stirring. After 5 h, the solid fraction was collected, washed with THF to remove excess of CDI, and then transferred to a new flask to which was added $N_3$/$NH_2$-UCNPs (1.25 mL of 14.5 mg/mL of NPs in THF). The slurry was gently stirred at room temperature for 24 h. The modified HCQ-$N_3$/$NH_2$-UCNPs were isolated via centrifugation and washed three times in water and finally suspended in 1 mL of water.

**Preparation of UCNPs: modification of HCQ-$N_3$/$NH_2$-UCNPs with PCL.** Initially, a solution of copper (II) sulphate pentahydrate (5.8 μL, 25 mM) was reduced to copper (I) by addition of sodium ascorbate (29 μL, 20 mM). This reduced copper (I) solution was added to a suspension of HCQ-$N_3$/$NH_2$-UCNPs (12 mg) in water/isopropanol (1.25 mL, 3:2, v/v) mixed with a solution of PCL (0.2 mg, 0.58 μmol, dissolved in 250 μL of isopropanol, Supplementary Figs. 21 and 22), and gently stirred for 1 h, in order to react the azide groups of HCQ-$N_3$/$NH_2$-UCNPs (Supplementary Fig. 23) with the alkyne group of PCL (Supplementary Fig. 24) via azide-alkyne cycloaddition. The PCL-HCQ-$N_3$/$NH_2$-UCNPs were isolated via centrifugation (6.000 rpm, 4629 g, 6 min) and washed three times in water and finally dispersed in water (500 μL).

**Preparation of UCNPs: loading of Cre onto the surface of the PCL-HCQ-$N_3$/$NH_2$-UCNPs.** PCL-HCQ-$N_3$/$NH_2$-UCNPs (1 mg) were dispersed in 100 mM MES buffer pH 6.4 (660 μL). A solution of N-hydroxysulfosuccinimide sodium salt (sulfo-NHS) and N-(3-dimethylaminopropyl)-N'-ethylcarbodiimide hydrochloride (EDC) was added (100 μL, 2.7 mM and 2.24 mM, respectively) and gently stirred for 15 min on an orbital shaker at the speed of 250 rpm. Cre recombinase (108 μg, 1.84 nmol) was gently added to the suspension and left to react for 20 h at 4°C. The Cre-PCL-HCQ-$N_3$/$NH_2$-UCNPs (from now on designated as Cre-UCNPs) were isolated via centrifugation (6.000 rpm, 3824 g, 6 min) and washed three times in 50 mM Tris-buffer pH 7.5 and finally dispersed in 50 mM Tris-buffer pH 7.5 (100 μL).

**Cell culture.** SC-1 mouse fetal embryo fibroblasts and the corresponding Cre reporter cell line transduced with the reporter allele SFr-II (from now on termed as Cre reporter fibroblasts)[64] (kindly donated by Dr. Carol Stocking, University of Hamburg, Germany) were grown in DMEM supplemented with fetal bovine serum (FBS, 10%) and PenStrep (0.5%), at 37 ˚C in a fully humidified air containing 5% $CO_2$. Both cells were passaged after reaching 80% confluency. The cell line has an expression cassette having the RFP gene in-frame to the ATG initiation codon with its adjacent loxP sequence (Fig. 3a). Translation of RFP is terminated at a UAG stop codon followed by a second loxP sequence. The ORF of a humanized eGFP, lacking an initiation codon, is downstream of the distal loxP site. Following Cre-mediated excision of RFP, eGFP codons are placed in-frame of the proximal ATG-LoxP sequence and are consequently translated.

**In vitro Cre-UCNP-mediated DNA recombination.** Cre reporter fibroblasts ($5 \times 10^3$ cells/well) were seeded onto a 96-well plate and left to adhere overnight. Cells were incubated with Cre-UCNPs (50 μg/mL), washed with PBS, and left to grow in complete medium for another 48 h. Cre-mediated DNA recombination was followed over time for a period of 48 h by direct analysis of GFP expression using the IN Cell Analyzer 2200 cell imaging system (GE Healthcare Life Sciences) equipped with an inverted 20x PlanFluor objective/0.45 numerical aperture. GFP expression was detected using an excitation filter 475/28 and an emission filter 511.5/23. The mean green intensity in the cell cytoplasm was quantified using the IN Cell Developer software. Flow cytometry was used to quantify the percentage of GFP-positive cells. For this purpose, cells after 40 h of Cre-UCNP incubation were dissociated with trypsin (0.1%, w/v, in PBS), centrifuged, resuspended in PBS and finally characterized by flow cytometry.

**NP cellular uptake by inductive coupled plasma mass spectrometry (ICP-MS) analyses.** Cre reporter fibroblasts ($5 \times 10^5$ cells/well) were plated in 6 well plates and left to adhere overnight. Then the cells were incubated with Cre-UCNPs (25 or 50 μg/mL) between 2 and 8 h, washed with PBS to remove noninternalized NPs, dissociated with trypsin (0.1%, w/v, in PBS), centrifuged and counted. The samples were freeze-dried, followed by the digestion with an aqueous nitric acid solution (9.9 mL, 2% w/v).

SVZ cells (see section *Animal testing: isolation of SVZ cells* described in Supplementary Information) were seeded in 24-well plates at a density of $10^5$ cells/well, whereas cortical neurons (see section *Preparation of neuronal cultures* described in Supplementary Information) were seeded in 12-well plates at a density of $1.8 \times 10^5$ cells/well. Cells were incubated with Cre-UCNPs (100, 200 and 300 μg/mL) for 4 h. After transfections, cells were washed with PBS to remove non-internalized NPs, dissociated with trypsin (0.1%, w/v, in PBS), centrifuged and counted. The samples were then freeze-dried, followed by the digestion with an aqueous nitric acid solution (9.9 mL, 2% w/v), and analyzed by ICP-MS for the quantification of internalized yttrium. The NP cellular uptake was also monitored in brain mice. Animals were subjected to a stereotaxic procedure for injection of Cre-UCNPs (30 μg) in the VTA (detailed protocol in section *Animal testing: VTA administration*). Immediately after injection, animals were perfused with PBS. The brains were excised, weighed and processed as before for ICP-MS analyses.

**NP cellular uptake and intracellular trafficking by confocal microscopy.** Intracellular trafficking of Cre-UCNPs was assessed on Cre-reported fibroblasts. Cells ($4 \times 10^4$ cells/well) were treated with Cre-UCNPs (50 μg/mL) with or without hydroxychloroquine for different time points: 5 min, 1 h, 4 h or 24 h. After

incubation, cells were washed extensively with PBS with the exception of one experimental group that was washed with culture medium and maintained for additional 24 h prior to fixation. Cells were fixed with 4% (v/v) paraformaldehyde for 15 min at room temperature and washed three times with PBS. Cells were permeabilized with 0.3% (w/v) Triton X-100. The detergent was removed by washing three times with PBS. Before staining, unspecific binding sites were blocked with 1% (w/v) BSA and 0.3 M glycine for 1 h. Cells were then stained for 1 h with primary antibodies (Supplementary Table 2): anti-Cre recombinase for the detection of Cre-UCNPs and their cargo (1:3000 MAB3120, Merck Milllipore), anti-EEA1 as a marker of early endosomes (1:100 clone: C45B10, Cell Signaling) and anti-LAMP1 (1:1000 ab24170, Abcam) to stain lysosomes. Primary antibodies were detected with an anti-mouse IgG Alexa Fluor 488 (Life Technologies) or an anti-rabbit IgG Cy3 conjugate (Jackson ImmunoResearch). Confocal fluorescent images were acquired using a Laser scanning confocal microscope (LSM 710, Zeiss), equipped with a 40x objective/1.4 numerical aperture oil PlanApochromat immersion lens. Alexa Fluor 488 was detected using the 488 nm laser line of an Ar laser (25 mW nominal output) and an LP 505 filter. The Cy3 signal was detected using a 561 nm HeNe laser (1 mW) and an LP 560 filter. The pinhole aperture was set to 1 Airy unit. Image acquisition was performed using the Zen Black 2012 software. Images were analyzed in ImageJ, and the colocalization was determined by calculating the Manders' colocalization coefficient between Cre-UCNPs and EEA1 or LAMP-1.

For studies with galectin-9, Cre-reported fibroblasts ($2×10^4$ cells/well in IBIDI μ-Slide 8 Well) were treated with Cre-UCNPs (25 μg/mL) with or without hydroxychloroquine for 4 h, then the cells were washed with culture medium and maintained for additional 20 h prior to fixation. As a positive control, cells were treated with soluble hydroxychloroquine (60 μM, 26 μg/mL) for 16 h. Cells were fixed with 4% (v/v) paraformaldehyde for 15 min at room temperature and washed three times with PBS. Cells were permeabilized with 0.3% (w/v) Triton X-100 for 15 min, and finally washed three times with PBS. Before staining, unspecific binding sites were blocked with 1% (w/v) BSA and 0.3 M glycine for 1 h. Cells were then incubated for 1 h with primary antibody rat anti-galectin-9 (1:100 cat. no. 137901, BioLegend). The antibody was removed by washing three times with PBS. The secondary antibody anti-rat IgG Alexa Fluor 488 (Life Technologies) were incubated by 1 h and the nuclei stained with DAPI (Sigma) for 5 min at RT. The galectin-9 foci were imaged using a Zeiss LSM 710 confocal microscope equipped with a Apochromat 40x/1.4 objective. The foci number and area were assessed by the Analyze Particles built-in function in FIJI.

For the studies with Lysotracker, cells ($4×10^4$ cells/well) were first incubated with Lysotracker Red (Life Technologies; 50 nM) for 30 min prior to Cre-UCNPs treatment. NPs with or without hydroxychloroquine (50 μg/mL) were exposed to cells for different time-points: 5 min, 10 min, 15 min and 1 h. After incubation, cells were washed extensively with PBS and paraformaldehyde (4% (v/v)). Cell membrane was stained with WGA (Invitrogen), nuclei stained with DAPI (Sigma), and the slides were mounted with mounting medium (Dako). Co-localization was examined using a Zeiss LSM 710 confocal microscope equipped with a Apochromat 40x/1.4 objective. The acquired images were analyzed by classical co-localization tools in FIJI. Vesicle number and area was assessed by the Analyze Particles built-in function in FIJI.

**NP cellular uptake pathways.** Cre reporter fibroblasts ($4×10^4$ cells/well) cells were seeded on a 24 well plate and left to adhere overnight. Cells were inhibited by one of the following chemicals during 30 min before adding a suspension of Dylight 488-labelled Cre-UCNPs: EIPA (100 μM), dynasor (5 μM), dansylcadaverine (10 μM), cytochalasin D (1 μM), nocodazole (3 μM), filipin III (100 μM) and polyinosinic acid (100 μg/mL). After 30 min, cells were incubated with Dylight 488-labelled Cre-UCNPs (25 μg/mL) for 4 h in the presence of the inhibitor. After 4 h, cells were washed one time with cold trypan blue solution, re-washed three times with cold PBS, dissociated with trypsin (0.1%, w/v, in PBS), centrifuged and resuspended in PBS for flow cytometry analyses. To validate the inhibitory activity of dynasor we performed uptake studies of FITC-labelled transferrin, known to selectively enter cells via clatherin-mediated endocytosis. Fibroblasts ($4×10^4$ cells/well) were seeded on a 24 well plate and treated or not with dynasor (30 min pre-incubation), followed by addition of FITC-labelled transferrin (1 μg/mL, Life Technologies). After 8 min, cells were evaluated as before by flow cytometry.

The NP uptake mechanism was evaluated on fibroblasts by silencing key regulators of CME (CLTC and LDLR), caveolin-mediated endocytosis (CAV1), GEEC-CCLIC pathways (CDC42), macropinocytosis (RAC1 and CTBP1) and scavenger receptors (SCARA3) with siRNAs (Thermo Fisher). Silencing was performed on 24-well plates with $4×10^4$ cells/well and confirmed by RT-qPCR. Total RNA was extracted using TRIzol (Invitrogen) and purified using the RNeasy Mini kit (Qiagen). Total RNA concentration and purity were analyzed using the NanoDrop spectrophotometer for quality control. First-strand cDNA was generated from 1 μg of RNA using the High Capacity cDNA Reverse Transcription kit (Applied Biosystems). The cDNA synthesis reaction followed a protocol of 10 min at 25 °C, followed by 2 h at 37 °C, and 5 min at 85 °C, executed by a CFX96 real-time PCR system (BioRad). Quantitative PCR was performed in the CFX96 RT-PCR system using the Power SYBR Green PCR Master Mix (Applied Biosystems). The reaction protocol started with an activation step at 50 °C for 2 min, followed by 2 min at 95 °C, and 40 cycles of amplification (denaturation at

95 °C for 15 s, then annealing and elongation at 60 °C for 1 min). Minimal cycle threshold (Ct) values for each gene were calculated from 3 technical replicates. Quantification of target genes (Supplementary Table 3) was performed using the Livak method ($2^{-ΔΔCt}$), comparing the ΔCt values obtained between each gene and the reference GAPDH gene to the negative control (No inhibitor).

Cells in antibiotic-free complete medium were transfected for 24 h with lipofectamine RNAiMAX (1.5 μL, Life Technologies) complexed with siRNA (100 nM). Next, cells were incubated with Dylight 488-labelled Cre-UCNPs (25 μg/mL) for 4 h, washed, with cold trypan blue solution, re-washed three times with cold PBS, dissociated with trypsin (0.1%, w/v, in PBS), centrifuged and resuspended in PBS for flow cytometry analyses. Non-transfected cells or cells transfected with lipofectamine complexed with scramble siRNA (MOCK) were used as controls.

**Animals.** R26YFP mice, in a C57BL/6 J background, were obtained as kind gift from Dr. Henrique Veiga-Fernandes (IMM, Lisbon). R26YFP mice were bred and maintained at Faculty of Medicine at the University of Coimbra animal facility. Male C57/BL6 and R26tdTomato mice were purchased from Charles River. All animals were housed at the animal house of the Faculty of Medicine at the University of Coimbra, in groups of 3 to 4 per cage, with food and water provided *ad libitum* and maintained in a 12 h light/dark cycle in temperature- and humidity-controlled rooms. For optogenetic studies, mice were single isolated after surgically intervened. When arriving, animals were acclimatized to the animal house and daily routines for at least one week after which they were handled twice a week. When needed, animal identification was performed by subcutaneous injection of green and/or black dyes in the paws. No randomization was used to allocate animals to experimental groups. All the animals used in the study were male with 7 to 20 weeks of age and were sacrificed by cervical dislocation at the end of each experiment.

**Animal testing: SVZ administration.** R26YFP mice (7 to 20-week-old) were anesthetized with isoflurane (1.5–2.0%), carprofen (1 mg/kg) via intraperitoneal injection and buprenorphine (0.1 mg/kg) via subcutaneous injection. Animals were placed on a heating pad maintained at 37 °C and allowed to reach a stable plane of anesthesia, which was periodically checked through examination of breathing rate and a toe-pinch test. The anesthetized mouse was then aligned on a Digital New Standard Stereotaxic frame (Stoelting), the skull was shaved, disinfected and craniotomies were performed with minimal damage to brain tissue. Mouse stereotaxic surgery target: sub ventricular zone (0.74 mm anterioposterior, +/− 0.75 mm mediolateral, −3, −2.7, −2.3 and −2 mm dorsoventral). All stereotaxic coordinates were relative to bregma. Cre-UCNPs (30 μg, 690 nL) were administered using a Nanoliter 2010 (World Precision Instruments).

**Animal testing: NIR activation of UCNPs.** Approximately 60 min after administrations, the injected area (above the craniotomy site) was irradiated with a NIR laser (980 nm; 425 mW/cm$^2$; 3 cycles of 5 min).

**Animal testing: preparation of implantable optical ferrules.** Custom-made implantable optical ferrules were constructed from 2.5 mm long 270 μm Ø ceramic stick ferrules (Thorlabs) based on a previously described design[65]. Briefly, a 200 μm Ø 0.39 NA optical fiber (Thorlabs) was stripped of its protective coating and cleaved. Next, cyanocrylate adhesive (Loctite) was applied to the concave end of the ferrule through which the cleaved fiber segment was threaded. After wiping off the excess glue, the ferrules were allowed to cure for 30 min at room temperature. The ferrule connector was polished using a polishing disk and increasingly fine grades of polishing paper (Thorlabs), with frequent inspection to ensure transmission quality. Once polished, the free end of the fiber was scored and cleaved to 5 mm in length.

**Animal testing: optical fiber implant.** For conditioned place preference experiments, mice were then implanted with implantable optical fibers, who were attached to the stereotaxic arm and tilted 10° outwards in the rostro-caudal axis. Two anchor stainless steel, sterile screws (M1.2) were implanted until contact with the dura and further fixated with VersaGlue PX-40 (Harvest Dental). Mouse stereotaxic surgery targets: ventral tegmental area (− 3.30 mm anterioposterior, + 0.50 mm mediolateral, −3.8 mm dorsoventral). The craniotomy window was kept well hydrated with 0.9% (w/v) NaCl throughout all the implant procedure. Dental cement (3 M) or Maxcem Elite (Kerr) was used to secure the optical implants in place. After verification that the implant is completely secured, a dental acrylic mixture was applied (Scheu Dental) to further secure the implant.

**Animal testing: VTA administration.** AAV5.CMV.PI.Cre.rBG (AAV5.Cre) and AAV5.EF1a.DIO.hChR2(H134R)-eYFP.WPRE.hGH (AAV5.DIO.ChR2.YFP) were acquired from Penn Vector Core. Male C57/BL6 mice (8 to 20-week-old) were anesthetized with isoflurane (1.5–2.0%), carprofen (1 mg/kg) via intraperitoneal injection, and buprenorphine (0.1 mg/kg) via subcutaneous injection. Animals were placed on a heating pad maintained at 37 °C and allowed to reach a stable plane of anesthesia, which was periodically checked through examination of breathing rate and a toe-pinch test. Their eyes were covered with Altalube ointment

(Altaire Pharmaceuticals) to prevent corneal drying. The anesthetized mouse was then aligned on a Digital New Standard Stereotaxic frame (Stoelting) and craniotomies were performed with minimal damage to brain tissue. All stereotaxic coordinates were relative to bregma.

For viral infusion, animals were injected with AAV5.DIO.ChR2.YFP alone (negative control) or AAV5.DIO.ChR2.YFP + Cre-UCNPs (30 μg, 690 nL) (test group) at the VTA (−3.3 mm anteroposterior, +0.5 mm mediolateral, −4.1, −4.0 and −3.9 mm dorsoventral). Animals were infused with a Nanoliter 2010 (total volume of 1.4 μL) tilted 10° outwards in the rostro-caudal axis. After infusion, the injector was left at the injection site for 10 min and then slowly withdrawn.

**Animal testing: behavioral analyses**. The custom-made conditioned place preference apparatus consisted of a rectangular cage with a left white chamber (17 cm × 12.5 cm) with a mesh grid floor, and a right black chamber (17 cm × 12.5 cm) with a sandstone floor grid. All behavioral video data was acquired with a Logitech C922 Pro Webcam and coded in Bonsai, a free open-source visual programming environment (https://bitbucket.org/horizongir/bonsai). The conditioned place preference test consisted of 5 days. Days 1 to 3 consisted of a preconditioning test that measured the time mice spent in each chamber to determine their preference (1 trial/day, 15 min/trial). On day 4, after 3 consecutive days with mice preferring one chamber instead of the other, the mice were allowed to freely move around the cage and were stimulated in the non-preferred side of the cage (trial duration: 30 min). On day 5, the mice were placed back into the chamber to assess preference for the stimulation and non-stimulation chambers (trial duration: 15 min).

**Animal testing: light stimulation for in vivo assays**. Optic fibers were attached through an FC/PC adaptor to a 465 nm LED module (Plexon), and light pulses were generated through a stimulator (Plexon). For all in vivo experimental protocols, 10 Hz blue light pulses over 3 min were delivered (S9). Optic fiber light intensity was measured at the tip using a light sensor (Thorlabs) and ranged from 5-6 mW.

**Animal testing: light stimulation for c-Fos induction**. Mice were anesthetized, 90 min after blue light stimulation, with chloral hydrate then perfused with 1× PBS followed by 4% (v/v) paraformaldehyde.

**Animal testing: intranasal administration**. R26tdTomato mice were subjected to intranasal administrations of the Cre-UCNPs (1 mg/mL) over the time of three days. The animals were anesthetized with 2.5% isoflurane, mice in the supine position were intranasally administered using a micropipette. The animals were administered with 40 μL of Cre-UCNPs equally divided for each nostril in each day. One hour after administration, the mice were anesthetized with isoflurane and one of the brain hemispheres was covered with aluminum foil before irradiating the other half with NIR light for 3 cycles of 5 min.

**Animal testing: immunohistochemistry**. Tissues were harvested and fixed overnight (or post-fixed for perfused samples) with 4% (v/v) paraformaldehyde then cryo-protected in sucrose [30% (w/v) in PBS]. Gastrocnemius muscles from the hind limb of mice were imbedded in OCT compound, snap-frozen and sectioned on a Cryostast (Leica) into 10 μm thick sections. Brains were sectioned (coronal plane sections) on a Cryostat (Leica) or on Vibrotome (Leica) into 50-μm thick sections. For Cryostat sectioning brains were snap-frozen before cutting. Sections were first treated with blocking solution (0.3% Triton X-100, 10% goat serum in PBS) and incubated with the primary antibody (Supplementary Table 2) in blocking solution for 8 h to 48 h at 4 °C. Sections were washed with 1× PBS and incubated with the secondary antibody at room temperature for 1 h to 24 h. Finally, sections were washed three times in 1× PBS, stained with the DNA binding fluorescence probe DAPI (1 μg/ml) and mounted on glass slides in a Prolong gold anti-fade medium (Thermo Fischer). Tile-scan pictures of the all section were obtained in the IN Cell Analyzer 2200 Cell Imaging System using 10x PlanApo objective/0.45 numerical aperture. A stitching program was done using FIJI software. Confocal fluorescent images were acquired using Laser scanning confocal microscope (LSM 710, Zeiss using an oil immersion Apochromat 40x/1.4 objective). Image analysis and quantification were done using FIJI software.

**Statistics & reproducibility**. Statistical analyses were performed with GraphPad Prism software (version 8). Statistical significance was analyzed using two-tailed unpaired Student's t test between two different groups. For multiple comparisons, a one-way ANOVA was performed followed by the appropriate multiple comparisons test, specified in the respective figure legend. Results were considered significant when p < 0.05. Data are shown as mean ± SEM unless other specification. No statistical method was used to predetermine sample size.

**Additional methods**. Further experimental details are available as Supplementary Methods in the Supplementary Information file.

**Reporting summary**. Further information on research design is available in the Nature Research Reporting Summary linked to this article.

## Data availability
The authors declare that the data supporting the findings of this study are available within the paper and its supplementary information files. All the figures have associated source data, which are provided with this paper. Source data are provided with this paper.

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

## Acknowledgements

L.F. acknowledges the funding by COMPETE2020 and FCT programs [POCI-01-0145-FEDER-016390 (acronym: CANCEL STEM), POCI-01-0145-FEDER-016682], Portugal 2020-COMPETE funding through "Programa Operacional Regional do Centro" CENTRO2020 (CENTRO-01-0145-FEDER-000014) and Projects Interreg 2IQBioNeuro ("Impulso de una red de I + i en química biológica para diagnóstico y tratamiento de enfermedades neurológicas") and NeuroAtlantic ("An Atlantic innovation platform on diagnosis and treatment of neurological diseases and aging", ref: EAPA_791/2018), and EC projects ERAatUC (Ref:669088) and RESETageing (Ref:952266). S.P. acknowledges the co-funding by the European Regional Development Fund (ERDF), through the Centro Regional Operational Program and by National Funds, through the Foundation for Science and Technology I.P.(FCT), CENTRO-01-0145-FEDER-028060 - PTDC/NAN-MAT/28060/2017 (acronym: BrainEdition), as well as FCT fellowships (SFRH/BPD/96048/2013). C.R. acknowledges the FCT fellowships (SFRH/BD/52337/2013). J.P. acknowledges funding from BIAL Foundation (BIAL 266/2016). A.F.R. is grateful to the EU Horizon 2020 programme for funding under grant agreement no. 101003413. The authors would like to thank Claudio Franco and Catarina Fonseca from IMM for the multiphoton microscopy studies.

## Author contributions

C.R., S.P. and L.F. designed the study, did the literature search and wrote the manuscript. C.R. conducted the study. C.R., S.P., T.R., C.S., A.F.R, S.S. and L.B collected the in vitro data. C.R., S.P., T.R., J.G., A.F.R, J.P and L.F. conducted and analyzed in vivo data.

## Competing interests

The authors declare the following competing interests: Lino Ferreira, Sonia Pinho and Catarina Rebelo are applicants of the patent entitled "Light triggerable gene editing formulation, in vivo editing method, and uses thereof", PCT/IB2020/054856. The patent described a formulation for in vivo gene editing with spatial control. The remaining authors declare no competing interests.

## Additional information

**Peer review information** *Nature Communications* thanks Guosong Hong, Balint Kiraly and other anonymous Reviewer(s) for the peer review of this work. Peer review Reports are available.

