## [Peer Review File · Nature Communications]

Reviewers' comments:

Reviewer #1 (Remarks to the Author):

Rebello et al. describe a method, in which near infrared light (NIR) induces photocleavage of Cre from upconverting nanoparticles (UCNPs), resulting in gene-editing controlled by NIR. They thoroughly demonstrate the method, with an impressive range of experiments. However, the promise of 'integrating gene editing with optogenetics' (see abstract) is not fulfilled, as optogenetics is not mediated by NIR and UCNPs, but is an entirely separate process by 'traditional' blue-light mediated applications. Also, the method is not novel since a similar application has been described by Morales et al. (2018). Therefore, the incremental improvement compared to the Morales paper as well as a lack of clear applications enabled by this method makes it unclear what conceptual advance the present study offers.

Major points

1. While the present manuscript sufficiently describes technical details, no clear applications enabled by this method are offered. The authors state that spatial control of gene editing by NIR is better compared to AAVs. However, this is not thoroughly demonstrated, and AAV-mediated expression can be controlled rather precisely in expert hands - definitely much better than what the authors show in Fig.6, and Fig.S18. Anatomical examples shown suggest widespread expression through multiple 'non-targeted' regions. Furthermore, the successful intranasal administration indicates that UCNPs are prone to spread to long distances, which does not support the view that UCNPs offer greater spatial resolution than AAVs.
2. A competing paper has been published in 2018. The authors only mention this in the one but last sentence (the Morales et al. citation is number 45 out of 45). This makes a false impression of novelty and does not provide fair credit to Morales et al. The more novel aspects (combination with NIR and UCNP-mediated optogenetic activation, non-invasive gene editing) should be examined and discussed in much greater depth.
3. Though the authors made significant improvements regarding the efficacy of gene editing with Cre-UCNPs compared to Morales et al., the new in vivo experimental results suggest that transgene expression did not meet the required level for successful non-invasive NIR-illumination-mediated ChR2-activation.
4. The in vivo experiments suffer from a number of problems. The expression in Fig. 4e is unconvincing. Instead of 'fluorescence intensity' in 'arbitrary units', proportion of expression neurons should be

provided as in Fig. c2. Non-expressing control with NIR illumination is missing from Fig. 6. Combined with the non-specific effects of NIR (see Fig. S16), this seems very important.

5. The strong concentration-dependence as shown e.g. in Fig. 4b, combined with the cytotoxicity demonstrated in Fig. S4-6 seem to strongly limit the applicability.

6. All experiments should have at least $n = 3$ (e.g. Fig. 4b, Fig. 6d).

Minor points:

line 281: 'ontogenetically'

line 293-295: The result of the described test is not presented in the text. Spatial resolution of the gene edition with this technique should be further discussed.

line 310: 'low s'

Reviewer #2 (Remarks to the Author):

In this work, Rebelo et al. report a near-infrared (NIR) activatable protein-conjugated nanoparticle for remotely controlled gene editing in the mouse brain. Specifically, this nanoparticle conjugate is composed of a NaYF₄:Tm,Yb@NaYF₄ upconversion nanoparticle (UCNP), Cre recombinase covalently conjugated to the UCNP via a photocleavable linker, and hydroxychloroquine that facilitates endolysosomal escape. The authors demonstrated that 980 nm NIR light can be converted to UV-to-blue light emission between 345 nm and 475 nm, thus effectively triggering the release of Cre recombinase and activating Cre-dependent gene editing. Using this system, the authors achieved gene editing in the subventricular zone (SVZ) and the ventral tegmental area (VTA) in vivo, with the latter example enabling Cre-dependent expression of ChR2 for optogenetic neuromodulation in a conditioned place preference (CPP) test. Lastly, the authors also demonstrated a noninvasive administration route to deliver this nanoparticle system via intranasal delivery. The most significant strength of this approach arises from its ability to use brain-penetrant NIR light for triggering gene editing in the deep mouse brain, thus removing a major roadblock, i.e., the need for fiber implantation, for spatiotemporally precise gene editing and optogenetics in vivo. Besides this major advantage, this paper also presents a wide array of in-vitro and in-vivo experiments to validate the necessity, sufficiency, efficacy, and utility of this novel approach. The authors are encouraged to improve the quality of data presentation by addressing the following questions:

- 1) The authors noted that Cre-UCNPs mediated gene editing exhibited higher efficiency than both soluble Cre (even with HCQ) and AAV transfection (e.g., AAV5-Cre). This difference can be seen from Fig. 3e. Although this nanoparticle-enhanced gene editing efficiency is interesting, the exact underlying mechanism remains unclear. For example, can nanoparticles with any chemical compositions facilitate highly efficacious gene editing, or is this efficacy specific to UCNPs? For example, can mesoporous silica nanoparticles be used to deliver Cre recombinase via physisorption and desorption?
- 2) The exact mechanism by which the HCQ molecule escapes the UCNP (shown in Fig. 1b) remains unclear. The amide bond should be sufficiently strong to prevent HCQ from detaching from the surface of UCNPs even in the presence of UV and blue light emission.
- 3) The demonstrated success of intranasal delivery of UCNP nanoparticles offers a noninvasive alternative for delivering these materials into the brain. Two mechanisms of intranasal drug delivery have been reported, intracellular (via transcytosis) and extracellular. Which mechanism was used by the UCNPs here?
- 4) The authors need to compare their 980-nm NIR triggered gene editing system with recently reported NIR-II triggered gene editing in vivo: PNAS 2020 117 (5) 2395-2405. The NIR-II window has demonstrated deeper brain penetration than the shorter-wavelength NIR counterpart: Nat. Photonics 2014, 8, 723-730.
- 5) Some claims in the manuscript are found to be unsubstantiated. For example, the authors state that “(Cre recombinase does) not offer spatial control over gene editing and coupling with optogenetics, and their functional output has not been demonstrated in brain cells”. Cre-dependent expression of opsins has been commonplace for neuroscience research: Nat. Neurosci. 2012, 15, 793-802. Region-specific expression of opsins can be achieved either via stereotaxic viral injection or crossing a cre-driver line with a cre-dependent reporter line.
- 6) Data presentation: Statistical analysis should be performed for pairwise comparison in Fig. 4b and Supplementary Fig. 3. In addition, Supplementary Fig. 5b.1 needs to have quantitative analysis, e.g., in a bar chart with statistical analysis. Furthermore, in Supplementary Fig. 18a.1, Gaussian fitting should be used to extract the size of Chr2 signal increases using AAVs and Cre-UCNPs. Is this difference in Chr2-expressing volume due to different diffusivity of AAVs vs Cre-UCNPs? If so, can the authors compare their diffusivity based on the Stokes-Einstein equation or experimental measurements with the difference in Chr2 expression volume here?
- 7) Experimental details: this reviewer found the reported concentrations of Cre recombinase inconsistent and confusing throughout the manuscript. For example, in Fig. 3 caption, Cre recombinase is reported as 0.48 $\mu\text{g}/\text{mL}$. However, in the graph of Fig. 3g, its concentration is reported in nM. Please be consistent with reported concentrations.
- 8) Minor issues: Fig. 3f and 3g are labeled incorrectly in the figure caption.

Reviewer #3 (Remarks to the Author):

In the manuscript by Rebelo et al., the authors describe the development and testing of new light-activated conditional recombination technology to afford the ability to deliver and activate Cre recombinase via the coupling to upconversion nanoparticles and near-infrared photoactivation. By testing the approach both in vitro and in vivo, the authors demonstrate high-efficiency recombination in either cells harboring floxed reporter constructs and/or in transgenic mice with embedded conditional reporter alleles. Overall, this is a novel approach with significant potential towards facilitating genetic modifications in a spatially-restricted and temporally-constrained manner, and using the NIR activation approach is different than other existing technologies. Unfortunately, in its current form the manuscript requires significant attention to further consider publication at Nature Communications. Attention to the raised concerns would significantly strengthen the work, and if they can be addressed the manuscript should be evaluated again.

Concerns/critiques:

-Figure 1a is not informative. This should either be removed and/or modified to present useful data towards the manuscript.

-Significant attention is placed on the utility of the hydroxychloroquine as a modification to avoid endolysosomal shuttling/degradation. Although data show differences in the efficacy of recombination w/wo HCQ, little is shown to support the proposed mechanism.

-Figure 2f displays data that are not compelling. It seems like the variance (although passing statistical significance) is broad enough to question efficacy. Even with HCQ, there is high levels of colocalization to LAMP1.

-No reference to Fig 2h in text, and it seems like Figs 3f and 3g are out of order.

-Unclear why the authors do not show the baseline RFP fluorescence in Fig 3 prior to recombination. This would strengthen the argument of efficiency with proper comparison.

-Cell death should be more fully considered for all in vitro analyses where numbers of recombined cells are analyzed. This might be facilitated by imaging the baseline RFP w/wo light recombination.

-Fig 4e-f require much more rigorous analysis of cell number and subtype. Given the YFP conditional reporter, it is unclear why the authors only use ROI fluorescence. It would be much stronger to actually count cells and determine what types of cells are labeled.

-Controls are lacking from data presented in 6e and f.

-The final descriptions of labeling through the nasal epithelium and olfactory bulb are ill-defined and nascent. If the authors choose to keep these data (currently only supplemental), they must elaborate. Otherwise, these data should just be removed.

-The text requires a thorough editing, given the numerous grammatical errors that detract from the science presented.

Reviewer #1:

Rebelo et al. describe a method, in which near infrared light (NIR) induces photocleavage of Cre from upconverting nanoparticles (UCNPs), resulting in gene-editing controlled by NIR. They thoroughly demonstrate the method, with an impressive range of experiments. However, the promise of ‘integrating gene edition with optogenetics’ (see abstract) is not fulfilled, as optogenetics is not mediated by NIR and UCNPs, but is an entirely separate process by ‘traditional’ blue-light mediated applications. Also, the method is not novel since a similar application has been described by Morales et al. (2018). Therefore, the incremental improvement compared to the Morales paper as well as a lack of clear applications enabled by this method makes it unclear what conceptual advance the present study offers.

The authors have revised the abstract of the manuscript in order to clarify the point raised by the reviewer with regards to the comment “*the promise of ‘integrating gene edition with optogenetics’ (see abstract) is not fulfilled, as optogenetics is not mediated by NIR and UCNPs, but is an entirely separate process by ‘traditional’ blue-light mediated applications.*”. In the revised version of the manuscript, the abstract was modified to: “Here, we report a protein-conjugated nanoformulation activated by near infrared (NIR) light that allows spatial control during in gene editing in deep brain tissue. We further demonstrate that the activity of the edited cells can be monitored by optogenetics using blue light as a trigger.”

With respect to the comment “*the method is not novel since a similar application has been described by Morales et al. (2018).*” The authors disagree with the reviewer. The method presented here is novel in the sense that explores a strategy to enhance endolysosomal escape of a protein-based gene editing system based on the immobilization of hydroxychloroquine (HCQ). In the revised version of the manuscript, the authors also provide new data about the mechanism of endosomal escape mediated by HCQ (see below). The study of Morales *et al.* uses Cre recombinase modified with a cell-penetrating peptide to overcome the lipophilic barrier of the endosomal membrane. They show the recombination of 21% and 24% of the cells using the nanoparticle system and a lipofectamine-based transfection agent, respectively. In our study, Cre-UCNP formulation is “125 times higher efficient in cell recombination than lipofectamine RNAiMax, and other transfection agents described for protein intracellular delivery such as cationic lipids²⁷, small transduction compounds after a hypertonic shock²⁸ or cell-penetrating peptide TAT dimers²⁹.” Therefore, the high capacity to deliver intracellularly proteins is itself a major contribution of our study for the field. Moreover, our contribution is not only in the method itself but also in demonstrating the possibility of adopting a novel non-invasive delivery strategy of a protein-based gene editing formulation into the brain (not yet reported). We also demonstrated the possibility to evaluate cell activity, by optogenetics, after gene editing. In the revised version of the manuscript the authors have added the following information (page 15): “The efficiency of our formulation relied on the incorporation of low amounts of HCQ chemically conjugated to the UCNPs (below 1 μ M) that induce endolysosomal escape. Previously, it has been shown that soluble chloroquine is one of the most effective agents to disrupt the endolysosomal compartment²⁵. The molecule induced the disruption of endolysosomes that were positive for late endosomal markers including CD62/Rab7 (~30%) and Rab9 (~23%), and the lysosomal marker LAMP1 (~45%)²⁵. Our results indicated that in the absence of light activation, Cre-UCNPs with HCQ, but not Cre-UCNPs without HCQ, efficiently escaped the endosomal compartment at levels similar to those of soluble HCQ. To demonstrate this escape, we have used galectin-9 protein as a reporter, since it has been shown to respond rapidly and with high sensitivity to endosomal membrane disruption²⁵. This protein is recruited to damaged endosomes during lipid NP or transfection agent-mediated endosomal escape²⁵. Our results showed that very small amounts of immobilized HCQ in NPs can be as effective as high concentrations of soluble HCQ (60-fold higher). It is possible that the immobilized HCQ molecules acted synergistically in the endosomal membrane, maximizing its effect. Importantly, we anticipate that the strategy described here to

facilitate endosomal escape based on immobilized HCQ can be easily incorporated into other delivery systems for protein or noncoding RNA delivery."

Regarding the comment "Therefore, (...) a lack of clear applications enabled by this method makes it unclear what conceptual advance the present study offers." Again, the authors disagree with the reviewer. The authors have demonstrated the potential of the formulation *in vivo* in three different paradigms (listed in the Abstract): "(i) gene editing in neurogenic niches, (ii) gene editing in the ventral tegmental area to facilitate precise optogenetic control of reward and reinforcement and (iii) gene editing with spatial control via a noninvasive administration route (i.e., intranasal)." Rather than focusing on a specific disease, our study offers multiple avenues to develop innovative therapies for neurological disorders. It should be noted that reward-pathway dysfunctions are very relevant in several disorders including psychiatric disorders (i.e., substance-use, affective, eating and obsessive compulsive), neurodevelopmental disorders (i.e. schizophrenia, attention-deficit/hyperactivity disorder, autism spectrum disorders) and genetic syndromes (i.e. Fragile X syndrome, Prader-Willi syndrome, Angelman syndrome, and Rett syndrome). These applications have been highlighted in the discussion section. In the revised version of the manuscript (page 14) we have added the following information: "We have demonstrated the utility of the formulation *in vivo* with the achievement of gene editing in neurogenic niches of the mouse brain, as well as the activation of mechanisms of reward and reinforcement in the VTA using NIR activation to genetically unlock an optogenetic actuator (ChR2), followed by its activation by blue light. The formulation reported here might be used in the treatment of reward-pathway dysfunctions that are highly relevant in psychiatric (e.g., substance-use, affective, eating and obsessive-compulsive disorders) and neurodevelopmental disorders (e.g. schizophrenia, attention-deficit/hyperactivity disorder and autism spectrum disorders), and genetic syndromes (i.e. fragile X syndrome, Prader-Willi syndrome, Angelman syndrome, and Rett syndrome)⁴⁶⁻⁴⁸."

Major points

1. While the present manuscript sufficiently describes technical details, no clear applications enabled by this method are offered. The authors state that spatial control of gene edition by NIR is better compared to AAVs. However, this is not thoroughly demonstrated, and AAV-mediated expression can be controlled rather precisely in expert hands - definitely much better than what the authors show in Fig.6, and Fig.S18. Anatomical examples shown suggest widespread expression through multiple 'non-targeted' regions. Furthermore, the successful intranasal administration indicates that UCNPs are prone to spread to long distances, which does not support the view that UCNPs offer greater spatial resolution than AAVs.

Regarding the comment: "While the present manuscript sufficiently describes technical details, no clear applications enabled by this method are offered", please see point above.

Regarding the comment: "The authors state that spatial control of gene edition by NIR is better compared to AAVs. However, this is not thoroughly demonstrated, and AAV-mediated expression can be controlled rather precisely in expert hands", the authors would like to clarify the reviewer that the results presented in Fig. S18 were obtained in the same administration conditions for both viral and Cre-UCNP formulations. In addition, although progresses have been reported in the development of AAV variants with enhanced tropism and transduction of brain cells (doi:[10.1016/j.tips.2021.03.004](https://doi.org/10.1016/j.tips.2021.03.004)), studies have shown that AAV serotypes such as AAV6 and AAV9 administered in the hippocampus by intracerebral administration can diffuse to brain regions at least 3 mm away from the injection site (in both directions) (<https://doi.org/10.1016/j.omtm.2019.06.005>; <https://doi.org/10.1038/mt.2008.166>). Moreover, striatal injections of AAV5 particles in brain mice mediated transduction in multiple regions located at least 4 mm away from the injection site (<https://www.jstor.org/stable/121909>). All these studies suggest that virus can diffuse to distant regions from the injection site although the transport mechanism is relatively unknown. In the case of Cre-UCNPs, the gene edition is determined by the spot area of the laser and thus the gene edition is delimited to the exposed area.

Regarding the comment: “Furthermore, the successful intranasal administration indicates that UCNPs are prone to spread to long distances, which does not support the view that UCNPs offer greater spatial resolution than AAVs.” the authors would like to stress that the spatial control triggered by Cre-UCNPs in gene edition was determined by the spot of the laser and not by their diffusion capacity. In the results obtained for Fig S19, only one brain hemisphere was irradiated while the opposite was protected from the laser. Therefore, the spatial resolution was at hemisphere level (the laser spot covered all hemisphere) and not at tissue level. In the near future, it might be possible to align the laser with a stereotaxic equipment to activate a particular region of the brain. Because the spatial resolution was at hemisphere level we could see gene edition in the cortex of the hemisphere that was irradiated. Figure S19 was revised to facilitate the reading. In the revised version of the manuscript, the following information has been incorporated (page 16): “The results reported here demonstrate our capacity to control the activity of genes with precise spatial control, using a Cre-based NP formulation that can be externally triggered using NIR light. (...) Second, we showed the capacity of Cre-UCNPs to induce *in vivo* cell recombination, with spatial control, at the hemisphere level (Supplementary Fig. 19). In this case, the spatial control in gene editing was controlled by the spot of the light and not defined by the point of injection (as demonstrated in the SVZ and VTA studies). In the near future, it might be possible to align the laser with a stereotaxic equipment to activate a particular region of the brain. For applications requiring deeper penetration of the light (more than 4 mm), a craniotomy (a clinical procedure that is used in humans to implant drug delivery systems in the brain⁵³) may be required. Third, our results further showed that the spatial control obtained by Cre-UCNPs in the VTA region was higher than that obtained with an AAV5 virus (Supplementary Fig. 18). After injection into the target area of the VTA, the AAV virus spread throughout the tissue with Chr2 expression appearing at distances of 3.5 mm from the injection site, while with the Cre-UCNPs, the expression of Chr2 was 1.5 mm from the injection site.”

2. A competing paper has been published in 2018. The authors only mention this in the one but last sentence (the Morales et al. citation is number 45 out of 45). This makes a false impression of novelty and does not provide fair credit to Morales et al. The more novel aspects (combination with NIR and UCNP-mediated optogenetic activation, non-invasive gene edition) should be examined and discussed in much greater depth.

Following the suggestions of the reviewer, the authors have revised the Introduction section to reference Morales study. In the revised version of the manuscript (page 3) the authors have added the following information: “Regulation of Cre recombinase formulations with light exposure has been shown to provide control over biological processes with an unprecedented precision. Light-activatable Cre recombinase systems have been based in split-Cre dimerization^{2, 3, 8-10}, the installation of a light-responsive o-nitrobenzyl caging group in the catalytic site of Cre¹¹ or via the fusion of a photocleavable protein to the Cre recombinase¹. However, these systems still display considerable limitations: (i) they are based on constitutive expression of Cre, and (ii) they rely on the activation using a UV/blue light, which has limited tissue penetration ability. A recent study has attempted to address these issues by developing a nanoparticle formulation capable of delivering Cre recombinase after activation by near infrared (NIR) light¹². However, this approach relied on the immobilization of Cre by noncovalent interactions with gold nanoshells, which resulted in poor control over protein delivery. In addition, the formulation showed limitations in terms of cellular uptake and intracellular delivery, which together translated into low cell recombination *in vitro* (approximately 20%), and its gene editing potential was not demonstrated *in vivo*.”

In addition, the authors have added additional information to the Introduction of the manuscript to highlight the importance of non-invasive gene edition (page 3): “Tackling complex biological questions, such as dissecting neuronal circuit function or stimulating regenerative processes in an intact organism, requires sophisticated technologies that facilitate precise genetic modification in live animals. Several approaches have been recently developed to control the activity of genes with precise spatiotemporal control, using enzymes such as Cre recombinase¹⁻³ or Cas9⁴⁻⁶ that can be

externally triggered by light. Nevertheless, despite the great progress made in recent years, our ability to controlling gene expression within biological tissue and simultaneously evaluate the activity of the edited cells by optogenetics has not been demonstrated. Viral strategies used to deliver gene-, or gene-modifying cargo into the mammalian brain are limited by several drawbacks including: (i) low biosafety profile of several commonly used vectors, (ii) limitations in cargo size capacity, and (iii) nonspecific infection of pre- and postsynaptic neurons at injection sites. Moreover, our capacity to deliver noninvasively gene-editing formulations into the brain is very limited. Most non-viral gene-editing formulations have been administered via an intracerebral route (reviewed in Ref. ⁷). Whether nonviral formulations can be administered through a non-intracerebral route and accumulate in the brain remains to be investigated. This strategy could enormously simplify the application of gene-editing formulations in the brain.

3. Though the authors made significant improvements regarding the efficacy of gene editing with Cre-UCNPs compared to Morales et al., the new in vivo experimental results suggest that transgene expression did not meet the required level for successful non-invasive NIR-illumination-mediated ChR2-activation.

The authors would like to clarify to the reviewer that the aim of the current work was not to demonstrate the non-invasive NIR-illumination-mediated ChR2-activation. Indeed, the concept of a NIR optogenetic antenna has been reported elsewhere (Chen *et al.*, Science 2018, 359, 679). The aim of the current work was to develop a light-responsive formulation that allows spatial control of gene expression in order to modulate cellular function and deconstruct the function of individual genes in biological processes. In a different note, the authors would like to clarify that the results presented in the current study do show transgene expression enough to have a functional impact. Indeed, the results in Figure 6 show that the ChR2 expression is enough to provide a functional readout. This experiment was only to prove functionality of the protein and not as an optogenetic antenna. Although the blue-upconverted light generated after NIR irradiation of UCNPs could potentially be used for inducing ChR2 activity, we were unable to demonstrate it due to the existence of other direct NIR-mediated membrane depolarizations (independent of ChR2 expression) in cortical neurons (Supplementary Fig. 16). Therefore, we used blue light to specifically dissect ChR2 activation in the VTA.

4. The in vivo experiments suffer from a number of problems. The expression in Fig. 4e is unconvincing. Instead of 'fluorescence intensity' in 'arbitrary units', proportion of expression neurons should be provided as in Fig. c2. Non-expressing control with NIR illumination is missing from Fig. 6. Combined with the non-specific effects of NIR (see Fig. S16), this seems very important.

Regarding the comment, "The expression in Fig. 4e is unconvincing", the authors wish to clarify that the endogenous expression of YFP after cell recombination is relatively weak. Thus, the authors have used an antibody against YFP to maximize the signal. These results are presented in Fig. 4e. The authors have removed from the figure the YFP channel without antibody staining.

Regarding the comment "Instead of "fluorescence intensity" in arbitrary units, proportion of expression neurons should be provided as in Fig. c2", in the revised version of the manuscript the authors have quantified the YFP- positive cells in the SVZ region and plotted the results in Fig. 4f.2. The authors did not characterize phenotypically the recombined cells because of the difficulty to separate the cells from each other in Y and Z axis using conventional characterization procedures. The quantification of phenotype of the edited cells would require unbiased stereological analyses which in turn would require the optimization of specific protocols to circumvent major issues with immunofluorescence, such as antigen retrieval to increase epitope availability, reduce collapse in tissue height during processing, and test other fluorophores to avoid photobleaching during data collection. Because of these technical hurdles, it was very difficult to collect the phenotype of the edited cells.

Regarding the comment “Non-expressing control with NIR illumination is missing from Fig. 6. Combined with the non-specific effects of NIR (see Fig. S16), this seems very important” the authors would like to clarify the reviewer that they wait 4 weeks until cell recombination and optogenetic tests were performed. The results in Fig. 6 were obtained after blue light activation and not by NIR activation. Therefore, the authors do not understand the relevance of the non-expressing control with NIR illumination.

5. The strong concentration-dependence as shown e.g. in Fig. 4b, combined with the cytotoxicity demonstrated in Fig. S4-6 seem to strongly limit the applicability.

In the revised version of the manuscript, the authors have removed one of the concentrations (100 µg/mL) because it was used for optimization processes in only two independent runs. The authors would like to clarify that the results presented in Supplemental figures 4, 5 and 6 were related to the cytotoxicity of Cre-UCNPs in fibroblasts which are cells that internalize higher amounts of nanoparticles than SVZ cells. The cytotoxicity results of Cre-UCNPs in SVZ cells was shown in Supplementary Fig.12. In this case, the effect of Cre-UCNPs in cell number and nuclear condensation (with DAPI staining) was evaluated. Our results indicate that the incubation of SVZ cells with Cre-UCNPs did not significantly increase the percentage of nuclear condensation (marker of cell death) up to 300 µg/mL. However, concentrations of Cre-UCNPs at 100 or 200 µg/mL affected to some extent SVZ cell proliferation, as evaluated by cell number counting after 4 days of NP exposure. This issue may not be particularly relevant *in vivo* since part of the cells at the SVZ niche are quiescent.

6. All experiments should have at least $n = 3$ (e.g. Fig. 4b, Fig. 6d).

The authors have revised the information in figures 4b and 6d. In figure 4b, the authors have removed one of the NP concentrations (with $n=2$) since it was used for optimization purposes (100 µg/mL). In Fig. 6d, the authors have re-adjusted the YY scale to show one of the points that was not shown in the previous version of the manuscript.

Minor points:

line 281: ‘ontogenetically’

line 293-295: The result of the described test is not presented in the text. Spatial resolution of the gene edition with this technique should be further discussed.

line 310: ‘low s’

The authors have performed the necessary corrections indicated by the reviewer in the text. Regarding the comment “Spatial resolution of the gene edition with this technique should be further discussed”, the authors have expanded the discussion section to clarify this issue. In the revised version of the manuscript, the authors have added the following information (page 16): “The results reported here demonstrate our capacity to control the activity of genes with precise spatial control, using a Cre-based NP formulation that can be externally triggered using NIR light. First, the capacity of Cre-UCNPs to induce *in vitro* cell recombination, with spatial resolution, was demonstrated at the single cell level (Fig. 3g). To the best of our knowledge, this level of control has not been demonstrated in the past using NIR light-triggerable NP formulations^{12, 51, 52}. Second, we showed the capacity of Cre-UCNPs to induce *in vivo* cell recombination, with spatial control, at the hemisphere level (Supplementary Fig. 19). In this case, the spatial control in gene editing was controlled by the spot of the light and not defined by the point of injection (as demonstrated in the SVZ and VTA studies). In the near future, it might be possible to align the laser with a stereotaxic equipment to activate a particular region of the brain. For applications requiring deeper penetration of the light (more than 4 mm), a craniotomy (a clinical procedure that is used in humans to implant drug delivery systems in the brain⁵³) may be required. Third, our results further showed that the spatial control obtained by Cre-UCNPs in the VTA region was higher than that obtained with an AAV5 virus (Supplementary Fig. 18). After injection into the target area of the VTA, the AAV virus spread throughout the tissue with ChR2 expression appearing at distances of 3.5 mm from the injection site, while with the Cre-UCNPs, the expression of ChR2 was 1.5 mm from the injection site.”

Reviewer #2:

1) The authors noted that Cre-UCNPs mediated gene editing exhibited higher efficiency than both soluble Cre (even with HCQ) and AAV transfection (e.g., AAV5-Cre). This difference can be seen from Fig. 3e. Although this nanoparticle-enhanced gene editing efficiency is interesting, the exact underlying mechanism remains unclear. For example, can nanoparticles with any chemical compositions facilitate highly efficacious gene editing, or is this efficacy specific to UCNPs? For example, can mesoporous silica nanoparticles be used to deliver Cre recombinase via physisorption and desorption?

The authors would like to thank the reviewer for the comment. In the initial submission, the authors have compared the performance of Cre-UCNPs relatively to RNAiMax transfection agent since it is commercially available and referenced in several gene editing studies (e.g. Zuris et al., Nature Biotechnology 2015, 33(1), 73). Others have used lipofectamine 2000 as a transfection agent (Wang et al., PNAS 2016, 113(11), 2868) which in your hands yielded poorer results than RNAiMax. In addition, in the revised version of the manuscript, the authors have included additional results to clarify the mechanism of Cre-UCNPs (Fig. 2g). In page 7, the authors have added the following information: “To evaluate the intracellular trafficking of the Cre-UCNPs, cells were exposed to Cre-UCNPs (50 µg/mL) and monitored by confocal microscopy (Fig. 2f) to determine the staining of Cre and lysosome-associated membrane protein 1 (LAMP1). After 5 min of exposure to NPs, the cells were characterized, and the co-localization of Cre-UCNPs with stained LAMP1 staining was reduced compared to that of the Cre-UCNPs without HCQ. Our results further showed that cells exposed to Cre-UCNPs with HCQ had a lower number of LAMP1-positive vesicles than those exposed to Cre-UCNPs without HCQ, which may indicate disrupted lysosome biogenesis (Supplementary Fig. 8). The decrease in the colocalization of Cre-UCNPs with LAMP1 may be explained by the fact that HCQ impairs autophagosome fusion with lysosomes²², and thus preventing the accumulation of NPs in LAMP1-positive vesicles, and/or by the disruption of the endolysosomal membrane compartment²⁵. To determine whether Cre-UCNPs with HCQ induced endolysosomal compartment disruption, we monitored galectin-9 by immunofluorescence, which is a very sensitive sensor of membrane damage and has been used to demonstrate endosomal escape of therapeutic molecules, including chloroquine²⁵. Indeed, the cells exposed to Cre-UCNPs with HCQ showed an increased number of galectin-9 foci compared to those treated with Cre-UCNPs without HCQ (Fig. 2g). Moreover, the large foci size found in the cells treated with Cre-UCNPs with HCQ further suggested the disruption of the endolysosomal compartment. These results also aligned with the results obtained by TEM analyses (Supplementary Fig. 7f). In contrast to Cre-UCNPs without HCQ, Cre-UCNPs with HCQ showed low colocalization within the endolysosomal compartment 4 h after transfection. Overall, our results indicated that Cre-UCNPs were taken up by cells mostly via clathrin-mediated endocytosis and that the NPs rapidly escaped the endolysosomal compartment due to the presence of HCQ.”

Finally, the authors would like to highlight that the immobilization of the Cre enzyme to any nanoparticle system by physisorption does not guarantee spatial control delivery of the gene editing system.

2) The exact mechanism by which the HCQ molecule escapes the UCNP (shown in Fig. 1b) remains unclear. The amide bond should be sufficiently strong to prevent HCQ from detaching from the surface of UCNPs even in the presence of UV and blue light emission.

The authors would like to thank the reviewer for the comment. In the revised version of the manuscript, the authors have included additional FTIR results (Supplementary Fig. 2d) that seems to indicate that HCQ is not released by the UCNPs after NIR exposure. Although the absorbance of HCQ is in a region of other molecular vibrations, there is no indication that the absorbance of HCQ has disappeared from UCNPs. Therefore, the mechanism of HCQ in favoring the intracellular delivery of Cre is mediated by the immobilized and not the soluble molecule. In the revised version of the manuscript, the authors have done additional experiments to further investigate the HCQ

mechanism. A study published in 2020 (Du Rietz *et al.*, Nature Communications 2020, 11, 1809) has shown that chloroquine disrupts the endolysosomal membrane which in turn leads to the appearance of galectin-9-positive foci. Thus, we have transfected fibroblast cells with Cre-UCNPs with and without HCQ and evaluate the number of intracellular galectin-9 foci. Importantly, these experiments were performed without NIR activation to highlight the endolysosomal properties of immobilized HCQ. Our results clearly show that cells transfected with Cre-UCNPs with HCQ have increased number of galectin-9-positive foci than cells transfected with Cre-UCNPs without HCQ. The authors have included these results in the revised version of the manuscript (see above). To the best of our knowledge, this is the first study demonstrating that immobilized HCQ contributes for endolysosomal escape using a very sensitive reporter system, which may open interesting opportunities in the context of intracellular delivery.

3) The demonstrated success of intranasal delivery of UCNP nanoparticles offers a noninvasive alternative for delivering these materials into the brain. Two mechanisms of intranasal drug delivery have been reported, intracellular (via transcytosis) and extracellular. Which mechanism was used by the UCNPs here?

Our results show that at early stages (within 4 h) Cre-UCNPs translocate from the nasal cavity and olfactory bulb to other parts of the brain such as the cortex and the hippocampus suggesting that extracellular transport is taking place. In the revised version of the manuscript, the authors have provided further details about the intranasal delivery mechanism (pages 12-13): “Despite the demonstration of *in vivo* gene editing and optogenetic actuation in freely moving animals, one of the main limitations of the presented strategy is the requirement of an invasive administration route. Therefore, we investigated the potential of using Cre-UCNPs to perform targeted gene editing through a non-invasive route, specifically intranasal instillation (Supplementary Fig. 19). Previous studies have demonstrated that nanomaterials administered through the nasal route can reach the brain^{43, 44}. Nose-to-brain transport has been reported for several nanomaterials, including exosomes⁴³ and iron oxide nanoparticles⁴⁴, among others, and can be primarily described by two mechanisms: intracellular (via transcytosis) and extracellular. The extracellular pathway is the faster route and includes the passage of the NPs by passive transport through tight junctions into the brain parenchyma⁴⁵. The intracellular pathway includes endocytosis of the NPs into olfactory and trigeminal nerve branches followed by axonal transport into brain. Initially, we investigated the distribution of the Cre-UCNPs after a single intranasal administration by ICP-MS analyses (4 h post-administration). Our results show that most of the NPs were in the nasal cavity, followed by the olfactory bulb (Supplementary Fig. 19a). Because UCre-UCNPs reached the cortex and hippocampus within a few hours, it is possible that the NPs reached these parts of the brain by extracellular transport. Next, we tested whether Cre-UCNPs induced gene editing with unilateral specificity by exposing only one of the brain hemispheres to NIR light. R26tdTomato mice were subjected to 3 intranasal administrations of Cre-UCNPs on 3 different days (i.e. days 1, 2 and 3). The animals were divided into two groups: the Cre-UCNP group received NPs (120 µg per administration), and the control group received only a saline solution. Each day, drops of NP suspensions were administered gradually into both nasal cavities, after which animals were subjected to NIR laser exposure in one of the brain hemispheres (the other was covered to prevent laser activation; 980 nm; 425 mW/cm²; 3 cycles of 5 min each) at 1 h after administration (Supplementary Fig. 19b). Our results showed recombined cells in the irradiated brain hemisphere, particularly in the olfactory bulb and cortex (Supplementary Fig. 19c). The area of tdTomato-positive cells in the olfactory bulb of the irradiated hemisphere was greater than that in the nonirradiated hemisphere (Supplementary Fig. 19d). Overall, our results demonstrated the utility of Cre-UCNPs for on-demand gene editing and the possibility to manipulate edited cellular circuits with optogenetic activation.”

4) The authors need to compare their 980-nm NIR triggered gene editing system with recently reported NIR-II triggered gene editing *in vivo*: PNAS 2020 117 (5) 2395-2405. The NIR-II window has demonstrated deeper brain penetration than the shorter-wavelength NIR counterpart: Nat. Photonics 2014, 8, 723-730.

In the revised version of the manuscript, the authors have discussed the advantages of the 980 nm NIT triggered gene editing system relatively to NIR-II. In page 18, the authors have included the following information: “During the preparation of this manuscript, NIR-activatable NPs able to release Cre recombinase with spatiotemporal control were described¹²; however, the formulation showed low gene editing (<20% of cell recombination), and the *in vivo* activity of the formulation was not demonstrated. Other recent studies have also documented the optical regulation of gene-editing formulations based on NIR. For example, an upconversion-activated CRISPR-Cas9 NP has been reported for the editing of cancer cells transplanted into skin tissue⁵¹. However, the study did not: (i) show the spatial resolution of the technology *in vitro* or *in vivo* (to evaluate the functional impact of their formulation, the authors injected the NPs in the tumor site, irradiated the tissue, and measured the reduction in tumor size), (ii) demonstrate gene editing with spatial resolution using a noninvasive route for the administration of the NPs and (iii) did not evaluate the cellular activity of the edited cells by optogenetics. Another study has reported a nanoparticle formulation for programmable genome editing in the second near-infrared (NIR-II) optical window⁵². The formulation was composed of a cationic polymer-coated gold nanorod and a Cas9 plasmid driven by the heat-inducible promoter HSP70. The authors have shown the capacity to regulate Cas9 activity by heat-induced gene expression. Although this system may show greater tissue-penetration than NPs that respond to the first near-infrared optical window (such as Cre-UCNPs), it suffered from some limitations such as (i) it was based in a plasmid and not in a protein (in contrast to the Cre-UCNPs), (ii) the long-term effects of the intracellular fluctuation of the temperature on cell activity remains to be determined, (iii) the transfection efficiency was 3-fold higher than that of lipofectamine, while in Cre-UCNPs, the transfection efficiency was more than 100-fold higher than that of a lipofectamine-based transfection agent (RNAiMax) and (iv) the study did not couple gene editing with optogenetics. Overall, the formulation reported here opens new avenues for *in vivo* spatial control of gene expression, which can be complemented with optogenetics to measure the activity of the edited cells.”

5) Some claims in the manuscript are found to be unsubstantiated. For example, the authors state that “(Cre recombinase does) not offer spatial control over gene edition and coupling with optogenetics, and their functional output has not been demonstrated in brain cells”. Cre-dependent expression of opsins has been commonplace for neuroscience research: Nat. Neurosci. 2012, 15, 793-802. Region-specific expression of opsins can be achieved either via stereotaxic viral injection or crossing a cre-driver line with a cre-dependent reporter line.

The points highlighted by the reviewer were for non-viral formulations. Anyway, the authors agree with the reviewer that they should include in this part of the work the viral formulations. Thus, in the revised version of the manuscript (page 9), the authors have added the following information: “Brain delivery of gene-editing protein formulations offers the potential to interrogate brain circuits and potentiate *in vivo* regenerative processes. Viral delivery of Cre-expressing constructs by stereotaxic administration into the brain or by crossing a cre-driver mouse line with a cre-dependent reporter mouse line has been performed; however, these may suffer limitations in terms of spatial resolution and biocompatibility issues (see below). Alternatively, *in vivo* gene editing of inner ear and brain cells has been demonstrated by delivery of Cre recombinase fused with a polyanionic protein and complexed with cationic lipids^{26, 30}, or by encapsulation in bioreducible lipid NPs²⁷. However, these formulations have limited transfection efficiency (see above), they require the fusion of Cre with a negatively charged protein, and they do not enable spatial control over gene editing.”

In addition, in the discussion section, the following information was added (page 15): “Our formulation may represent a combined alternative with reduced safety problems in the rhodopsin delivery strategy and efficient NIR-mediated stimulation. A widely used approach to study gene function *in vivo* is Cre/loxP recombination through viral delivery of Cre-expressing constructs. However, even though the widespread use of this technique is an indication of its usefulness, several studies have reported nonspecific and potentially noxious effects of Cre recombinase in mice in various organs, including the brain^{55, 56}. In addition, a recent study demonstrated that local AAV-mediated Cre expression in the substantia nigra of wild-type mice induced a massive decrease in neuronal populations, greatly impacting the nigrostriatal dopaminergic pathway, culminating in an

anatomical and functional phenotype resembling models of dopaminergic degeneration⁵⁷. Another method to achieve Cre/loxP recombination is through Cre-driver mouse lines, where optogenetic tools, delivered via Cre-inducible viral constructs, are expressed only in defined neurons expressing Cre recombinase. However, the use of these mouse lines has drawbacks. For example, nonspecific expression of Cre recombinase may occur outside the target cell or brain region due to promiscuous activity of the driving promoter^{58, 59}. In addition, Cre-driver lines can present inherent physiological problems, such as altered metabolic phenotypes, impacting the body length and body weight of the mice⁶⁰. These findings underline the necessity for in-depth scrutiny of Cre toxicity and spatial localization in the brain and further affirm our formulation as a possible tool to mitigate the side effects of existing techniques.”

6) Data presentation: Statistical analysis should be performed for pairwise comparison in Fig. 4b and Supplementary Fig. 3. In addition, Supplementary Fig. 5b.1 needs to have quantitative analysis, e.g., in a bar chart with statistical analysis. Furthermore, in Supplementary Fig. 18a.1, Gaussian fitting should be used to extract the size of ChR2 signal increases using AAVs and Cre-UCNPs. Is this difference in ChR2-expressing volume due to different diffusivity of AAVs vs Cre-UCNPs? If so, can the authors compare their diffusivity based on the Stokes-Einstein equation or experimental measurements with the difference in ChR2 expression volume here?

In the revised version of the manuscript, the authors have included the statistical analyses in Fig. 4b, Supplementary Fig. 3 and Supplementary Fig. 5b.2 (the results of Fig. 5b.1 are plotted in Fig. 5b.2; no statistical differences were observed among the groups as mentioned in the caption). The authors have also done Gaussian fitting to extract the size of ChR2 signal increase in AAV and Cre-UCNP conditions. We replaced the previous Supplementary Fig. 18a by a new plot. It is possible that the differences in ChR2-expressing volume are due to differences in the diffusivity of AAVs and Cre-UCNPs. Indeed, previous studies indicate that AAV migration in the brain is very influenced by its serotype (DOI: <https://doi.org/10.1016/j.omtm.2019.06.005>; <https://doi.org/10.1073/pnas.97.7.3428>). We would love to perform the diffusivity calculations suggested by the reviewer but we are not sure how we can do those calculations since we do not have information about some parameters such as medium viscosity and virus physical properties.

7) Experimental details: this reviewer found the reported concentrations of Cre recombinase inconsistent and confusing throughout the manuscript. For example, in Fig. 3 caption, Cre recombinase is reported as 0.48 µg/mL. However, in the graph of Fig. 3g, its concentration is reported in nM. Please be consistent with reported concentrations.

The authors would like to thank the reviewer for the note. The authors have plotted the concentration units in Figure 3f in nM, in order to benchmark against commercial transfection agents and other NP-based systems reported in the literature. We have clarified in the main text: “In our studies, 2.5 nM of Cre (i.e. 10 µg/mL Cre-UCNP) were sufficient to achieve recombination in 50% of cells”.

We also corrected the caption of Figure 3 to: “In Cre-UCNPs (50 µg/mL) the concentrations of immobilized Cre and HCQ were 12.5 nM (0.48 µg/mL) and 0.22 µM, respectively. Therefore, as a control, cells were treated with a solution containing the same concentrations of Cre and HCQ.”

8) Minor issues: Fig. 3f and 3g are labeled incorrectly in the figure caption.

In the revised version of the manuscript, the authors have correctly labeled the figure captions.

Reviewer #3:

- 1) Figure 1a is not informative. This should either be removed and/or modified to present useful data towards the manuscript.

The authors have removed Fig.1a according to the suggestion of the reviewer.

- 2) Significant attention is placed on the utility of the hydroxychloroquine as a modification to avoid endolysosomal shuttling/degradation. Although data show differences in the efficacy of recombination w/wo HCQ, little is shown to support the proposed mechanism.

In the previous version of the manuscript the authors have shown that Cre-UCNPs with HCQ had lower accumulation in lysosome (LAMP1-positive organelles) compartment than Cre-UCNPs without HCQ. Thus, this indicates that Cre-UCNPs might be more efficient in escaping the endolysosomal compartment. In the revised version of the manuscript, the authors have done additional experiments to further investigate the HCQ mechanism. A study published in 2020 (Du Rietz *et al.*, Nature Communications 2020, 11, 1809) has shown that chloroquine disrupts the endolysosomal membrane which in turn leads to the appearance of galectin-9-positive foci. Thus, we have transfected fibroblast cells with Cre-UCNPs with and without HCQ and evaluate the number of intracellular galectin-9 foci. Our results show that cells transfected with Cre-UCNPs with HCQ have increased number of galectin-9-positive foci than cells transfected with Cre-UCNPs without HCQ, confirming that Cre-UCNPs with HCQ are more effective in disrupting the endolysosomal compartment. The authors have included the following information in page 7 of the revised version of the manuscript: “To evaluate the intracellular trafficking of the Cre-UCNPs, cells were exposed to Cre-UCNPs (50 µg/mL) and monitored by confocal microscopy (Fig. 2f) to determine the staining of Cre and lysosome-associated membrane protein 1 (LAMP1). After 5 min of exposure to NPs, the cells were characterized, and the co-localization of Cre-UCNPs with stained LAMP1 staining was reduced compared to that of the Cre-UCNPs without HCQ. Our results further showed that cells exposed to Cre-UCNPs with HCQ had a lower number of LAMP1-positive vesicles than those exposed to Cre-UCNPs without HCQ, which may indicate disrupted lysosome biogenesis (Supplementary Fig. 8). The decrease in the colocalization of Cre-UCNPs with LAMP1 may be explained by the fact that HCQ impairs autophagosome fusion with lysosomes²², and thus preventing the accumulation of NPs in LAMP1-positive vesicles, and/or by the disruption of the endolysosomal membrane compartment²⁵. To determine whether Cre-UCNPs with HCQ induced endolysosomal compartment disruption, we monitored galectin-9 by immunofluorescence, which is a very sensitive sensor of membrane damage and has been used to demonstrate endosomal escape of therapeutic molecules, including chloroquine²⁵. Indeed, the cells exposed to Cre-UCNPs with HCQ showed an increased number of galectin-9 foci compared to those treated with Cre-UCNPs without HCQ (Fig. 2g). Moreover, the large foci size found in the cells treated with Cre-UCNPs with HCQ further suggested the disruption of the endolysosomal compartment. These results also aligned with the results obtained by TEM analyses (Supplementary Fig. 7f). In contrast to Cre-UCNPs without HCQ, Cre-UCNPs with HCQ showed low colocalization within the endolysosomal compartment 4 h after transfection. Overall, our results indicated that Cre-UCNPs were taken up by cells mostly via clathrin-mediated endocytosis and that the NPs rapidly escaped the endolysosomal compartment due to the presence of HCQ.”

- 3) Figure 2f displays data that are not compelling. It seems like the variance (although passing statistical significance) is broad enough to question efficacy. Even with HCQ, there is high levels of colocalization to LAMP1.

The authors have performed additional experiments (Fig.2g) to show the mechanism of immobilized HCQ. Please see point 2.

- 4) No reference to Fig 2h in text, and it seems like Figs 3f and 3g are out of order.

The authors would like the reviewer for the note. In the revised version of the manuscript, the authors have changed the order of Fig.3f and Fig.3g. In addition, they have referenced Fig. 2h (in the revised

version of the manuscript the authors moved it to Supplementary information as Supplementary figure 7f.2).

- 5) Unclear why the authors do not show the baseline RFP fluorescence in Fig 3 prior to recombination. This would strengthen the argument of efficiency with proper comparison.

In the previous version of the manuscript, the authors have shown RFP fluorescence levels during cell recombination by flow cytometry analyses (please see Fig. 3d). Now, in the revised version of the manuscript, the authors have also included fluorescence images to illustrate the maintenance of RFP levels during cell recombination (Supplementary Fig. 9).

- 6) Cell death should be more fully considered for all *in vitro* analyses where numbers of recombined cells are analyzed. This might be facilitated by imaging the baseline RFP w/wo light recombination.

The authors would like to stress that the viability of recombined cells was previously shown in Supplementary Figs. 4d-4f. Cre-UCNPs (at concentrations up to 50 $\mu\text{g}/\text{mL}$) had no major effect in cell survival as measured by cell counting (48 h after the activation of the recombination process) (Supplementary Fig. 4d), Annexin/PI analyses (Supplementary Fig. 4e) and ATP measurements (Supplementary Fig. 4f).

- 7) Fig 4e-f require much more rigorous analysis of cell number and subtype. Given the YFP conditional reporter, it is unclear why the authors only use ROI fluorescence. It would be much stronger to actually count cells and determine what types of cells are labeled.

In the revised version of the manuscript the authors have quantified the YFP-positive cells in the SVZ region and plotted the results in Fig. 4f.2. The authors did not characterize phenotypically the recombined cells because of the difficulty to separate the cells from each other in Y and Z axis using conventional characterization procedures. The quantification of phenotype of the edited cells would require unbiased stereological analyses which in turn would require the optimization of specific protocols to circumvent major issues with immunofluorescence, such as antigen retrieval to increase epitope availability, reduce collapse in tissue height during processing, and test other fluorophores to avoid photobleaching during data collection. Because of these technical hurdles, it was very difficult to collect the phenotype of the edited cells.

- 8) Controls are lacking from data presented in 6e and f.

The authors are not sure if the reviewer is asking for a control about the transfection of brain tissue with AAV5.DIO.ChR2.YFP followed by Cre-UCNPs without laser activation. We have not conducted this control because the results obtained in Figure 6d were clear about the absence of activity in this experimental group. In addition, the research conducted in this paper was performed in accordance to European (Directive 2010/63/EU) and Portuguese law (Decreto-Lei n.º 113/2013) where the rational use of laboratory animals is key, with strict and demanding ethical standards, based on the adherence to the principle of the 3Rs, with the firm objective of using the least number of animals, in the best possible conditions.

- 9) The final descriptions of labeling through the nasal epithelium and olfactory bulb are ill-defined and nascent. If the authors choose to keep these data (currently only supplemental), they must elaborate. Otherwise, these data should just be removed.

The authors would like to thank the reviewer for the comment. The authors have removed some data to make this section more focused. In addition, the authors have elaborated more about the experiment performed. In the revised version of the manuscript the authors have included the following information: "Despite the demonstration of *in vivo* gene editing and optogenetic actuation in freely moving animals, one of the main limitations of the presented strategy is the requirement of

an invasive administration route. Therefore, we investigated the potential of using Cre-UCNPs to perform targeted gene editing through a non-invasive route, specifically intranasal instillation (Supplementary Fig. 19). Previous studies have demonstrated that nanomaterials administered through the nasal route can reach the brain^{43, 44}. Nose-to-brain transport has been reported for several nanomaterials, including exosomes⁴³ and iron oxide nanoparticles⁴⁴, among others, and can be primarily described by two mechanisms: intracellular (via transcytosis) and extracellular. The extracellular pathway is the faster route and includes the passage of the NPs by passive transport through tight junctions into the brain parenchyma⁴⁵. The intracellular pathway includes endocytosis of the NPs into olfactory and trigeminal nerve branches followed by axonal transport into brain. Initially, we investigated the distribution of the Cre-UCNPs after a single intranasal administration by ICP-MS analyses (4 h post-administration). Our results show that most of the NPs were in the nasal cavity, followed by the olfactory bulb (Supplementary Fig. 19a). Because UCre-UCNPs reached the cortex and hippocampus within a few hours, it is possible that the NPs reached these parts of the brain by extracellular transport. Next, we tested whether Cre-UCNPs induced gene editing with unilateral specificity by exposing only one of the brain hemispheres to NIR light. R26tdTomato mice were subjected to 3 intranasal administrations of Cre-UCNPs on 3 different days (i.e. days 1, 2 and 3). The animals were divided into two groups: the Cre-UCNP group received NPs (120 µg per administration), and the control group received only a saline solution. Each day, drops of NP suspensions were administered gradually into both nasal cavities, after which animals were subjected to NIR laser exposure in one of the brain hemispheres (the other was covered to prevent laser activation; 980 nm; 425 mW/cm²; 3 cycles of 5 min each) at 1 h after administration (Supplementary Fig. 19b). Our results showed recombined cells in the irradiated brain hemisphere, particularly in the olfactory bulb and cortex (Supplementary Fig. 19c). The area of tdTomato-positive cells in the olfactory bulb of the irradiated hemisphere was greater than that in the nonirradiated hemisphere (Supplementary Fig. 19d). Overall, our results demonstrated the utility of Cre-UCNPs for on-demand gene editing and the possibility to manipulate edited cellular circuits with optogenetic activation.”

10) The text requires a thorough editing, given the numerous grammatical errors that detract from the science presented.

The authors have revised the manuscript following the suggestion of the reviewer.

REVIEWERS' COMMENTS

Reviewer #1 (Remarks to the Author):

Unfortunately, the manuscript by Rebelo et al. has remained largely unchanged. Thus, I am still on the opinion that the described method seems largely technically solid (although I am not a NIR expert), but I do not see clear applications with great advantage. The abstract still stresses 'spatial control', but then the response to the points mentions that 'spatial resolution was at hemisphere level', which does not suggest a strong 'spatial control'. In another point of the rebuttal letter, the Authors stress the relevance to psychiatric disorders. If the Authors consider this direction the main novel application, then their manuscript should focus on this and convince the reader about this specific advance. Nevertheless, I leave it to the Editor's discretion whether she finds the method promising enough that merits publication in Nature Communications, as I find this rather subjective and more a matter of journal preference than a scientific question.

I believe that openness and transparency can increase the fairness of peer review. Therefore, I decided to sign my reviews.

Balazs Hangya

Reviewer #2 (Remarks to the Author):

The authors have addressed my comments with satisfaction. I hereby recommend acceptance of this work for publication in Nature Communications.

Reviewer #3 (Remarks to the Author):

I am satisfied with the author responses to the previous reviews and now endorse publication per other reviewer suggestions.

Reviewer #1

“Unfortunately, the manuscript by Rebelo et al. has remained largely unchanged. Thus, I am still on the opinion that the described method seems largely technically solid (although I am not a NIR expert), but I do not see clear applications with great advantage. The abstract still stresses ‘spatial control’, but then the response to the points mentions that ‘spatial resolution was at hemisphere level’, which does not suggest a strong ‘spatial control’. In another point of the rebuttal letter, the Authors stress the relevance to psychiatric disorders. If the Authors consider this direction the main novel application, then their manuscript should focus on this and convince the reader about this specific advance. Nevertheless, I leave it to the Editor’s discretion whether she finds the method promising enough that merits publication in Nature Communications, as I find this rather subjective and more a matter of journal preference than a scientific question.

I believe that openness and transparency can increase the fairness of peer review. Therefore, I decided to sign my reviews.

Balazs Hangya”

In the revised version of the manuscript, the authors have tune down the claims of spatial control. The abstract has been reformulated to: “Here, we report a formulation composed of upconversion nanoparticles (UCNPs) conjugated with Cre recombinase enzyme through a photocleavable linker and a lysosomotropic agent that facilitates endolysosomal escape. This formulation allows *in vitro* spatial control in gene editing after activation with near-infrared light. We further demonstrate the potential of this formulation *in vivo* through three different paradigms: (i) gene editing in neurogenic niches, (ii) gene editing in the ventral tegmental area to facilitate monitoring of edited cells by precise optogenetic control of reward and reinforcement, and (iii) gene editing in a localized brain region via a noninvasive administration route (i.e., intranasal).”

Reviewer #2

The authors have addressed my comments with satisfaction. I hereby recommend acceptance of this work for publication in Nature Communications.

The authors would like to thank the reviewer for the positive assessment of the work.

Reviewer #3

I am satisfied with the author responses to the previous reviews and now endorse publication per other reviewer suggestions.

The authors would like to thank the reviewer for the positive assessment of the work.